# Curious Causality-Seeking Agents in Open-ended Worlds

**Zhiyu Zhao**[1], **Haoxuan Li**[2], **Haifeng Zhang**[3,4], **Jun Wang**[5], **Francesco Faccio**[6,7],
**Jürgen Schmidhuber**[6,7] **Mengyue Yang**[1]*
[1]University of Bristol    [2]Peking University
[3]Institute of Automation, Chinese Academy of Sciences
[4]School of Artificial Intelligence, Chinese Academy of Sciences
[5] University College London
[6]King Abdullah University of Science and Technology
[7]The Swiss AI Lab, IDSIA-USI/SUPSI

## Abstract

When building a world model, a common assumption is that the environment has a single, unchanging underlying causal rule, like applying Newton's laws to every situation. However, in truly open-ended environments, the apparent causal mechanism may drift over time because the agent continually encounters novel contexts and operates within a limited observational window. This brings about a problem that, when building a world model, even subtle shifts in policy or environment states can alter the very observed causal mechanisms. In this work, we introduce the **Meta-Causal Graph** as world models for open-ended environments, a minimal unified representation that efficiently encodes the transformation rules governing how causal structures shift across different latent world states. A single Meta-Causal Graph is composed of multiple causal subgraphs, each triggered by meta state, which is in the latent state space. Building on this representation, we introduce a **Causality-Seeking Agent** whose objectives are to (1) identify the meta states that trigger each subgraph, (2) discover the corresponding causal relationships by agent curiosity-driven intervention policy, and (3) iteratively refine the Meta-Causal Graph through ongoing curiosity-driven exploration and agent experiences. Experiments on both synthetic tasks and a challenging robot arm manipulation task demonstrate that our method robustly captures shifts in causal dynamics and generalizes effectively to previously unseen contexts.

## 1  Introduction

World models [21, 62, 70] have emerged as a critical component in reinforcement learning, enabling agents to simulate and plan in complex environments. These models aim to capture the underlying dynamics of the environment, allowing agents to predict future states and evaluate potential actions [23, 21, 62, 55]. However, simply learning which variables correlate can mislead an agent when the world's dynamics change. Recently, causality has been adopted to improve world models for interactive and complex environments, since causal rules—describing how one factor brings about another—capture the underlying data-generation process and yield more robust and generalizable decision-making [36, 59]. While many existing approaches incorporate notions of causality, they typically fail to capture the open-world causal relationships.

To identify the underlying world rules, traditional causal models rely predominantly on observational data [46], implicitly assuming a fixed causal structure. In open-ended environments, although the

---

*Corresponding author: mengyue.yang@bristol.ac.uk

39th Conference on Neural Information Processing Systems (NeurIPS 2025).

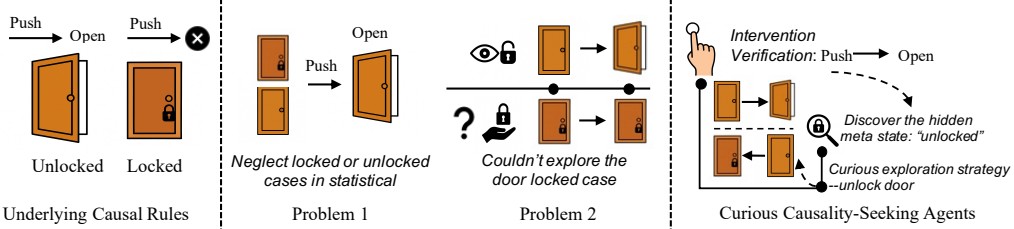

Figure 1: Illustration of the Meta-Causal Graph concept and the Curious Causality-Seeking Agent framework. **Ground Truth**: The causal relationship between pushing and opening depends on the latent state (locked vs. unlocked). **Limitations of Existing Approaches**: Problem 1: Single uniform causal graphs fail to capture context-dependent variations in causal relationships; Problem 2: Domain modeling requires a priori knowledge of state labels, limiting generalization to novel contexts. **Our Approach**: Curious Causality-Seeking Agents actively intervene to verify causal relationships and discover the critical meta states that determine when causal structures change, enabling the agent to build a comprehensive Meta-Causal Graph without requiring predefined domain labels.

ground-truth causal laws remain unchanged, the observational data is collected through local, context-limited exploration. Under conditions of partial observability, this invariance becomes obscured within a limited observational window, giving rise to the illusion of shifting causal relationships—a rule valid in one context may break down in another. For instance, the "push $\rightarrow$ open" relation holds for an unlocked door, but this relation vanishes when the door is locked. In this case, the same action produces no effect, leaving the true causal mechanism ambiguous. Traditional causal modeling in such scenarios faces two main challenges: (1) methods that learn a single uniform (time series-based) causal graph neglect these context-dependent changes [47], and (2) lack of an active curiosity-driven exploration strategy, which makes it struggle to adapt to discover the global causal rules in rich, unseen variations of potential open-ended worlds [29, 31].

To deal with these two challenges, we propose the **Meta-Causal Graph** to describe the world, which is a unified structure that captures how causal relationships evolve across different states. Rather than relying on a uniform graph, the Meta-Causal Graph is a minimal representation that contains multiple causal subgraphs, each corresponding to a particular meta state. Transitions between these meta states "trigger" the activation of the appropriate subgraph, allowing us to model shifts in causal influence, such as when a locked door severs the "push $\rightarrow$ open" link. We introduce a framework termed **Curious Causality-Seeking Agent**, which actively explores interventions to uncover the Meta-Causal Graph. The main contributions of this work are given below:

Theoretically, we derive sufficient conditions ensuring that strategically chosen families of interventions uniquely identify meta-causal graphs, overcoming observational limitations.

Methodologically, we design a framework which leverages a curiosity-driven exploration strategy where agents selectively intervene in the environment to learn a causal world model. First, we employ interventional verification to directly test whether specific variables causally influence others under different state conditions. Unlike methods relying solely on observational data, our approach actively intervenes on variables to establish causal links. Second, we introduce a targeted exploration strategy designed to discover meta states, critical configurations where causal structure changes.

Empirically, our approach demonstrates substantial improvements over purely observational baselines in simulated benchmarks, showcasing its capability to accurately recover complex causal structures. By explicitly modeling and reasoning about these causal relationships through active interventions, our proposed method significantly enhances the robustness and adaptability of world models.

## 2  Preliminaries

In this work, we employ causal modeling to construct world models. We begin by introducing fundamental definitions and terminology related to causal graphs and interventions.

**Causal Graph.** A causal subgraph is formalized as a directed acyclic graph (DAG) $\mathcal{G} = ([p], E)$, where the vertex set $[p] = \{1, 2, \ldots, p\}$ indexes a collection of random variables $X = \{X_i\}_{i=1}^{p}$ with

joint density $f(X)$. The edge set $E \subseteq [p] \times [p]$ encodes direct causal relationships between variables. For each node $i \in [p]$, we write $\text{Pa}_{\mathcal{G}}(i)$ for its set of parents. An edge $i \to j \in E$ indicates that $X_i$ exerts a direct causal influence on $X_j$.

**Intervention.**   Directly learning the causal structure from observational data is often insufficient, as it can lead to multiple possible causal graphs that share the same v-structures. To resolve this ambiguity, we can use interventions to break the observational ambiguity. Intervention is a powerful tool in causality. An intervention refers to the manipulation of a variable within a causal model, often represented by the do-operator. For example, consider intervening on state $X_k$, where the system originally takes the value $X_k = x_k$. An intervention is performed by setting the state variable to a new value, as in $\text{do}(X_i = x_i')$, such as setting "door open" to true in the environment.

During world-model learning, we perform a sequence of environment interventions to reveal causal structure. To better formalize the intervention influence, in this paper, we introduce the notion of an **intervention target**. At intervention step $k$, we select a single variable $X_i$ as the target, which we denote by the intervention target $I_k = \{i\}$, where $i \in [p]$. Applying an intervention $\text{do}(X_{I_k} = x')$ removes all incoming causal links into $X_i$. Concretely, if $\mathcal{G}$ is the original DAG, the *intervention graph* $\mathcal{G}^{(k)}$ is obtained by removing every edge of the form $j \to i, \quad \forall j \in \text{Pa}_{\mathcal{G}}(i)$, from $\mathcal{G}$. This modified graph accurately reflects the altered generative process under the do-operation on $X_i$.

**Definition 1** (Intervention graph [27]). *Let $D = ([p], E)$ be a DAG with vertex set $[p]$ and edge set $E$, and $I \subset [p]$ an intervention target. The intervention graph of $D$ is the DAG $D^{(I)} = ([p], E^{(I)})$, where $E^{(I)} := \{a \to b \mid a \to b \in E, b \notin I\}$.*

Given $\mathcal{I}$, the **Interventional Markov Equivalence** is defined as follows:

**Definition 2** (Interventional Markov Equivalence [27]). *Two DAGs $D_1$ and $D_2$ are interventional Markov equivalent given a set of intervention target $\mathcal{I}$ if $D_1$ and $D_2$ have the same skeleton and the same v-structures, and $D_1^{(I)}$ and $D_2^{(I)}$ have the same skeleton and the same v-structures for all $I \in \mathcal{I}$.*

Then the **Interventional Markov Equivalence Class (I-MEC)** is the set of DAGs that are interventional Markov equivalent to each other given a set of intervention targets $\mathcal{I}$. Given a set of intervention targets $\mathcal{I}$, we can determine the I-MEC of the system. However, the DAGs in the I-MEC are not unique. We denote the set of DAGs in the I-MEC as $\mathcal{D}_{\mathcal{I}}$. Although DAGs in $\mathcal{D}_{\mathcal{I}}$ share the same skeleton and v-structures, they may contain edges with different directions.

## 3   Meta-Causal Graph: Definition and Identifiability

In this section, we present our Meta-Causal Graph for a world formulation. We begin by defining the Meta-Causal Graph, which is a compact causal graph comprising multiple context-specific causal subgraphs, varying from latent meta states in Section 3.1. Then, we show the identifiability of meta states in Section 3.2 and their indicated causal subgraphs in Section 3.3 by interventions.

### 3.1   Meta-Causal Graph

We consider an environment whose states are represented by a set of variables $X = \{X_i\}_{i \in [p]} \in \mathcal{X}$, such that the "door open" is a state in the environment. $X$ in the environment can be intervened on by manipulating its values directly or influenced by taking actions indirectly. Then, the Meta-Causal Graph is defined on the environment state as follows:

**Definition 3** (Meta-Causal Graph). *The Meta-Causal Graph $\mathcal{MG}$ consists of a collection of causal subgraphs $\mathcal{MG} = \{\mathcal{G}_u\}_{u \in U}$, where each $\mathcal{G}_u$ corresponds to a distinct meta state $u \in U$. The causality skeleton matrix $M_u$ is a binary matrix where $M_u[i,j] = 1$ indicates that variable $i$ is a parent of variable $j$ (i.e., $i \in Pa(j)$) in the causal graph $\mathcal{G}_u$. A state-to-meta state mapping $C : \mathcal{X} \to U$ assigns each system observation $\mathbf{x} \in \mathcal{X}$ to its active meta state $u = C(\mathbf{x})$. We call $\mathcal{MG} = (\{\mathcal{G}_u\}, C)$ a Meta-Causal Graph if the following conditions are satisfied:*

*1. There exists a ground truth mapping $C : \mathcal{X} \to U$ determining the real meta states.*

*2. Causality skeleton matrices of causal subgraphs are sufficiently different. For all $u \neq u'$, $M_u \neq M_{u'}$, i.e., there exists an index $(i,j)$ such that $M_u[i,j] \neq M_{u'}[i,j]$.*

The underlying Meta-Causal Graph in the environment is assumed to be invariant; however, its complete identification requires two steps: (1) identification of latent meta states, and (2) identification of the causal subgraph corresponding to each meta state.

## 3.2 Identifiability of Meta States

In this subsection, we will discuss the identifiability of the meta states from the environment. We first introduce the concept of *swap-label equivalence* between two mappings.

**Definition 4** (Swap-Label Equivalence). *Two mappings $C_1 : \mathcal{X} \to U_1$ and $C_2 : \mathcal{X} \to U_2$ are swap-label equivalent if there exists a permutation $g : U_1 \to U_2$ such that $C_2(x) = g(C_1(x)), \forall x \in \mathcal{X}$.*

The definition indicates that if we can identify the meta states up to an permutation transformation, we say that the meta states are identifiable up to label swapping. We then show that the meta states are identifiable up to swap-label equivalence under the following assumption.

**Assumption 1** (Mixed Data Structure Learning). *Let $\mathcal{MG} = \{\mathcal{G}_i\}_K$ be a Meta-Causal Graph with $K$ causal subgraphs corresponding to distinct meta states $u \in U$. Consider a dataset $\mathcal{D}$ where each sample is generated from one of these causal subgraphs. Let $S_\mathcal{D} \subseteq \{1, 2, ..., K\}$ denote the indices of causal subgraphs that actually contributed samples to $\mathcal{D}$. When learning a single causal graph $\hat{\mathcal{G}}$ from the mixed dataset $\mathcal{D}$ (treating all samples as if they were generated from a single causal subgraph), the estimated parent set for each variable $X_j$ in $\hat{\mathcal{G}}$ satisfies:*

$$Pa_{\hat{\mathcal{G}}}(X_j) = \bigcup_{i \in S_\mathcal{D}} Pa_{\mathcal{G}_i}(X_j) \quad \forall j \in [p]. \tag{1}$$

Under this assumption, the unified graph from pooled data contains the union of all direct parent–child edges present in each causal subgraph. Consequently, the latent meta state are identifiable as follows.

**Theorem 1** (Identifiability of Meta States). *Under Assumption 1, the learned mapping $\hat{C} : \mathcal{X} \to \hat{U}$ is swap-label equivalent to the ground truth mapping $C : \mathcal{X} \to U$.*

**Identifiability when number of meta states is unknown:** In practice, we often overparameterize the meta state space by choosing more clusters than actually exist. We prove that this overparameterization does not harm identifiability: even with excess clusters, each true causal subgraph can still be recovered. The following theorem shows that, provided the number of clusters is sufficiently large, the underlying subgraph structures are identifiable up to the observational equivalence.

**Definition 5.** *Two mappings $C_1 : \mathcal{X} \to U_1$ and $C_2 : \mathcal{X} \to U_2$ are observationally equivalent if there exists a mapping $\phi : U_1 \to U_2$ such that $C_2(x) = \phi(C_1(x))$ for all $x \in \mathcal{X}$.*

Intuitively, observational equivalence refers to the condition where two mappings from states to meta states produce the same partitioning of observations, possibly up to relabeling.

**Theorem 2** (Identifiability of Overparameterized Meta States). *Under Assumption 1, if the learned mapping $\hat{C} : \mathcal{X} \to \hat{U}$ satisfies $|\hat{U}| > |U|$, the learned mapping $\hat{C}$ is observationally equivalent to the ground truth mapping $C : \mathcal{X} \to U$.*

## 3.3 Identification of Causal Subgraphs

In this subsection, we demonstrate that causal subgraphs for each meta state become uniquely identifiable once edge directionality is determined through interventions.

### 3.3.1 Causal Subgraph Identification

We employ multiple interventions to determine the causal structure of environment, represented as **intervention target indicator set** $\mathcal{I} = \{I_k\}_{k=1}^K$, where $I_k \subset \{[p]\}$ comprises the indices of intervened variables selected from state variable set $X = \{X_i\}_{i \in [p]}$ at intervention step $k$. These intervention sets are collected from historical data or agent interactions. We establish sufficient conditions ensuring causal graph edge identifiability through appropriate interventional selection.

**Proposition 1** (Identifiability of Causal Subgraph). *The causal subgraph $D$ can be uniquely identified if there exists an intervention target indicator set $\mathcal{I}$ such that for all edges $a \to b \in D$, there exists an intervention target $I \in \mathcal{I}$ such that $|I \cap \{a, b\}| = 1$.*

This proposition establishes that causal subgraph identifiability requires edge-specific interventions targeting single connected variables, enabling directional determination.

# 4 Curious Causality-Seeking Agents for Open-Ended World Modeling

We propose a novel framework to learn the Meta-Causal Graph through a Causality-Seeking Agent. Our approach comprises three core components: (1) curiosity-driven interventional exploration in an open-ended world, (2) Meta-Causal Graph discovery from agent experience, and (3) continual world model learning and updating. Together, these components enable the agent to actively discover and refine causal structures in open-ended environments.

## 4.1 Curiosity-Driven Interventional Exploration in Open-Ended Worlds

Accurate identification of the Meta-Causal Graph cannot rely solely on passive experience. First, causal structures inferred from purely observational trajectories often suffer from edge misorientation. Second, an agent's experience typically spans only a limited subset of world states, leaving parts of the causal graph unobserved. To reveal the full Meta-Causal Graph in an open-ended environment, the agent must complement passive observation with targeted, curiosity-driven interventions that actively probe uncertain causal relations.

**Curiosity Reward.** Our framework treats curiosity as a general intrinsic signal that can be instantiated in multiple forms. Specifically, we experimented with the following formulations:

*(1) Edge-Entropy.* $I_t^{\text{edge}} = \sum_{i,j \in [p]} H\big(\hat{M}_{C(X_t)}[i,j]\big)$, where $\hat{M}_{C(X_t)}[i,j] \in [0,1]$ denotes the posterior probability of an edge from variable $i$ to $j$ in the current meta state $C(X_t)$, and $H(\cdot)$ is the Shannon entropy. This term prioritizes interventions on the most uncertain parts of the causal graph.

*(2) Prediction Uncertainty.* $\mathcal{I}_t^{\text{unc}} = H\big[p_\theta(x_{t+1} \mid x_t, a_t)\big]$, where $p_\theta(x_{t+1} \mid x_t, a_t)$ is the world model's predictive distribution over the next state given current state–action pair $(x_t, a_t)$. Higher entropy indicates epistemic uncertainty about future transitions.

*(3) Feature Discrepancy.* $\mathcal{I}_t^{\text{feat}} = \big\| E(x_{t+1}) - E_\theta(\hat{x}_{t+1}) \big\|_2^2$, where $E(\cdot)$ is the learned feature encoder and $E_\theta(\hat{x}_{t+1})$ is the predicted feature of the next state $\hat{x}_{t+1}$. This term measures reconstruction error in the feature space, highlighting dynamics that the model fails to capture.

*(4) Predictive-Distribution Discrepancy.* $\mathcal{I}_t^{\text{nll}} = -\log p_\theta(x_{t+1} \mid x_t, a_t)$, which is the surprisal or negative log-likelihood of the observed transition.

Any of these intrinsic terms can be used as the curiosity reward $R_t$ to select interventions, and in our main results we adopt the edge-entropy variant as the default while other forms yield comparable behaviors (see Table 9).

**Intervention Verification.** After executing curiosity-guided interventions, we validate and refine the learned causal structures by directly estimating causal effects. For each variable $X_i$, we perform interventions $\text{do}(X_i = x_i')$ and observe the resulting changes in other variables. We estimate the causal effect of $X_i$ on $X_j$ as

$$\Delta_{ij} = \log P(X_j^{t+1} \mid X^t, \text{do}(X_i = x_i')) - \log P(X_j^{t+1} \mid X^t).$$

We integrate these effects into a mask-refinement loss

$$\mathcal{L}_{\text{mask}} = -\lambda_1 \sum_{\{(i,j):|\Delta_{ij}|>\tau\}} \log \hat{M}_{ij} + \lambda_2 \sum_{\{(i,j):|\Delta_{ij}|<\tau\}} \log \hat{M}_{ij},$$

where $\tau \geq 0$ controls sensitivity.

**Interventional Reachability.** In realistic environments, not all variables are directly intervenable. To formalize which causal relations are identifiable under such constraints, we introduce *interventional reachability*. We define an *intervention operator* $F \in \{0,1\}^{N \times N}$ where $F[k,i] = 1$ if and only if state $k$ can be obtained from state $i$ by a single allowed intervention, and a *transition operator* $T \in \{0,1\}^{N \times N}$ for the environment's natural dynamics after intervention. The composite $TF$

models one intervention–transition cycle, and nonzero entries of $(TF)^k z_0$ enumerate states reachable within $k$ cycles. A state is reachable if and only if some finite $k$ satisfies $[(TF)^k z_0][i] > 0$. These constraints define the feasible set of causal interventions. Details are given in Appendix E.

## 4.2 Meta-Causal Graph Discovery from Agent Experience

Given trajectories collected through curiosity-driven interventions, the agent infers a structured representation of causal dependencies that vary across latent meta states. The goal is to jointly learn (i) the discrete latent variable $C(X_t)$ indicating the current meta state, and (ii) the corresponding causal skeleton matrix $M_{C(X_t)}$ that governs the local dynamics.

**Representation Learning.** To model context-dependent causal dependencies from trajectories generated by curiosity-driven interventions, we discretize the latent dynamics of the environment using a vector-quantized variational auto-encoder (VQ-VAE) [31]. The encoder maps concatenated state–action pairs $(x_t, a_t)$ into a latent vector $z_e$, which is quantized to the nearest codebook entry $z_u$ associated with a meta state $u$. Each codebook entry defines a distribution over causal structures, from which the causal skeleton matrix $M_{C(X_t)}$ is sampled. Further details are provided in Appendix E.

**Effect of Lossy Representation.** The latent representation learned through vector quantization is inherently lossy due to imperfect mapping. Such lossy representations may distort the underlying causal factors and thus influence the identification of both meta states and their associated causal subgraphs. We provide an error bound of lossy representation in the following.

**Proposition 2** (Effect of Representation Accuracy on Misclassification). *Given a set of ground-truth meta states $U = \{u\}$ and corresponding codebook entry index $\hat{U} = \{\hat{u}_k\}$, let $p_k = \sum_{u \in U} \mu_u p_k^u$ denote the probability that a sample $x$ is mapped to codebook entry $z_{\hat{u}_k}$. Then, under the lossy representation induced by encoder, the probability of misclassification is*

$$P(\text{misclassification}) = 1 - \sum_k \sum_{u \in U} \mu_u p_k^u (1 - p_k + p_k^u \mu_u)^n.$$

*When $n(p_k + p_k^u \mu_u) \ll 1$, this can be approximated as*

$$P(\text{misclassification}) \approx \sum_k \left[ p_k^2 - \sum_u (\mu_u p_k^u)^2 \right].$$

**Adaptive Codebook Fusion.** To prevent redundant meta states resulted from overparamterization and maintain a minimal representation, we adopt an *Adaptive Codebook Fusion* mechanism. During training, codebook entries encoding similar latent embeddings and producing comparable decoded causal skeleton matrices are automatically merged. This fusion step is integrated into the quantization update, requires no extra supervision, and reallocates capacity for novel meta states.

## 4.3 Transition Probability Learning

The previous subsections describe how to identify the parent set of each state variable, $Pa_{\mathcal{G}_u}(X_j)$, using the learned causal skeleton matrix $M_\theta$. In this section, we demonstrate how to leverage the confirmed causal skeleton to quantify inter-variable causal effects, thereby grounding and validating the transition probabilities of the world model for next-state prediction. We train our model by maximizing the log-likelihood of the observed sequences, thereby learning the transition dynamics.

$$\mathcal{L}_{\text{MLE}} = -\log P(X_j^{t+1} | Pa_{\mathcal{G}_u}(X_j)).$$

**The complete world model learning objective.** To encourage the discovery of parsimonious causal structures that capture essential relationships while avoiding spurious connections, we incorporate a sparsity regularization term: $\mathcal{L}_{\text{sparse}} = \|\hat{M}_\theta\|_1$, where $\|\hat{M}_\theta\|_1$ denotes the L1 norm of the causal skeleton probability matrix, promoting sparsity in the learned causal graphs. The complete optimization objective integrates multiple components is as follows:

$$\mathcal{L} = \mathcal{L}_{\text{MLE}} + \lambda_{\text{sparse}} \mathcal{L}_{\text{sparse}} + \lambda_{\text{mask}} \mathcal{L}_{\text{mask}} + \lambda_{\text{quantization}} \mathcal{L}_{\text{quantization}}.$$

Training proceeds by alternating among interventional exploration, causal subgraph updating, world model learning, and codebook refinement. Algorithm 1 summarizes the complete learning procedure.

Table 1: Prediction accuracy on OOD states in the Chemical environment. The number of noisy nodes in the downstream tasks is denoted as $n$. All values are reported in percentage (%).

| Algorithm | Fork | | | Chain | | |
|---|---|---|---|---|---|---|
| | n=2 | n=4 | n=6 | n=2 | n=4 | n=6 |
| GNN | 36.29±3.45 | 25.80±3.48 | 21.58±3.44 | 29.22±3.39 | 23.28±4.98 | 20.53±6.96 |
| MLP | 31.11±1.69 | 30.44±2.28 | 32.39±1.76 | 28.66±3.65 | 26.52±4.26 | 24.15±4.17 |
| NCD | 41.60±5.08 | 37.47±2.13 | 42.27±1.82 | 40.04±6.21 | 37.47±2.98 | 41.19±1.66 |
| FCDL | 57.82±9.90 | 49.29±8.90 | 47.70±6.68 | 50.66±10.10 | 48.81±8.91 | 48.05±5.86 |
| Modular | 26.53±3.45 | 24.73±5.61 | 26.73±8.31 | 25.24±4.68 | 24.94±4.81 | 25.09±5.91 |
| NPS | 40.56±4.61 | 26.81±4.37 | 23.02±4.27 | 38.73±2.63 | 27.69±4.28 | 24.45±3.84 |
| CDL | 35.59±1.85 | 35.82±1.40 | 42.22±1.39 | 34.90±1.59 | 36.52±1.72 | 42.06±1.29 |
| GRADER | 37.93±1.06 | 38.94±1.63 | 45.74±2.25 | 36.82±3.12 | 37.41±2.84 | 43.48±4.14 |
| Oracle | 33.87±1.34 | 36.48±1.80 | 42.47±0.75 | 34.63±1.78 | 38.31±2.48 | 42.87±2.08 |
| Sandy-Mixure | 31.93±0.16 | 32.47±0.0161 | 33.72±2.11 | 30.08±3.12 | 29.31±5.18 | 27.43±2.20 |
| Transformer | 25.13±0.63 | 24.37±2.58 | 21.90±2.35 | 29.62±0.65 | 30.37±2.81 | 29.78±1.08 |
| MCG (wo mask loss) | 58.18±14.51 | 51.70±8.58 | 46.04±8.56 | 47.75±7.52 | 46.19±5.87 | 47.94±6.60 |
| MCG (wo intervention verification) | 48.28±8.15 | 43.48±5.05 | 46.54±3.72 | 50.36±8.05 | **50.55±6.83** | 48.72±4.42 |
| **MCG (ours)** | **63.18±13.94** | **50.47±9.87** | **50.04±8.56** | **51.99±6.58** | 49.78±4.11 | **49.69±5.14** |

Table 2: Average episode rewards on downstream tasks for each environment. The number of noisy nodes introduced in the downstream tasks is denoted as $n$.

| Algorithm | Chain | | | Fork | | | Magnetic |
|---|---|---|---|---|---|---|---|
| | n=2 | n=4 | n=6 | n=2 | n=4 | n=6 | |
| GNN | 6.89±0.28 | 6.38±0.28 | 6.56±0.53 | 6.61±0.92 | 6.15±0.74 | 6.95±0.78 | 2.23 ± 0.90 |
| MLP | 7.39±0.65 | 6.63±0.58 | 6.78±0.93 | 6.49±0.48 | 5.93±0.71 | 6.84±1.17 | 2.10 ± 0.22 |
| NCD | 9.60±1.52 | 8.86±0.23 | 10.32±0.37 | 10.95±1.63 | 9.11±0.63 | 9.11±0.63 | 2.85 ± 0.47 |
| FCDL | 11.16±3.5 | 10.39±2.84 | 10.62±2.52 | 13.98±2.01 | 13.36±2.14 | 12.91±2.40 | 2.77 ± 0.45 |
| Modular | 6.61±0.63 | 7.01±0.55 | 7.04±1.07 | 6.05±0.70 | 5.65±0.50 | 6.43±1.00 | 0.88 ± 0.52 |
| NPS | 6.92±1.03 | 6.88±0.79 | 6.80±0.39 | 5.82±0.83 | 5.75±0.57 | 5.54±0.80 | 0.91 ± 0.69 |
| CDL | 8.71±0.55 | 8.65±0.38 | 10.23±0.50 | 9.37±1.33 | 8.23±0.40 | 9.50±1.18 | 1.10 ± 0.67 |
| Oracle | 8.47±0.69 | 8.85±0.78 | 10.29±0.37 | 7.83±0.87 | 8.04±0.62 | 9.66±0.21 | 0.95 ± 0.55 |
| Sandy-Mixture | 6.81±0.17 | 6.73±0.20 | 7.07±0.26 | 6.95±0.21 | 6.71±0.20 | 7.03±0.40 | 1.63 ± 0.02 |
| Transformer | 6.45±0.28 | 6.73±0.18 | 7.31±0.33 | 6.54±0.18 | 6.68±0.27 | 7.00±0.39 | 2.13 ± 0.01 |
| **MCG(ours)** | **13.82±3.84** | **12.49±2.39** | **12.45±1.37** | **14.65±2.75** | **14.06±2.64** | **13.28±2.04** | **3.19 ± 0.14** |

## 5 Related Work

**World Models.** World models [21, 62, 70] enable agents to summarize past interactions, predict future states in the environment, and evaluate candidate actions [23, 21, 55, 26, 25]. However, methods that rely solely on statistical correlations often break down when environmental conditions shift, undermining their ability to generalize robustly [59, 77, 18, 19].

**Causal Discovery for Open-Ended World.** Causal discovery provides a rigorous framework for modeling the generative processes that govern complex systems, revealing how one variable brings about changes in another. By explicitly representing cause–effect relationships, causal methods yield more compact and invariant descriptions of reality than do purely statistical correlations [36, 44, 81, 5, 8, 9, 15, 45]. Causal representation learning provides a useful tool to learn causal structure, which aims to recover causally meaningful latent factors and structures, enabling invariance and transfer in downstream tasks [64, 48, 39, 80, 79, 78]. However, most of the work of causal representation learning focusses on learning a static causal structures passively from observational data.

However, identifying causal structure from observational data alone is a challenge since multiple directed acyclic graphs can entail the same conditional independencies and v-structures [76]. To resolve these ambiguities, researchers have turned to the collection of active causal interventional data by actively perturbing variables to distinguish among candidate graphs [52, 33, 72, 6, 66]. More recent work has even addressed settings with unknown intervention targets [27, 28, 14, 41].

Despite these advances, most causal discovery algorithms presume a single uniform, context-independent causal graph [47, 2, 43, 71, 83, 67, 82, 7] in the world. Some recent methods have sought to learn causal structures in changing world [57, 10, 75, 58, 49, 32, 31, 34], but they typically

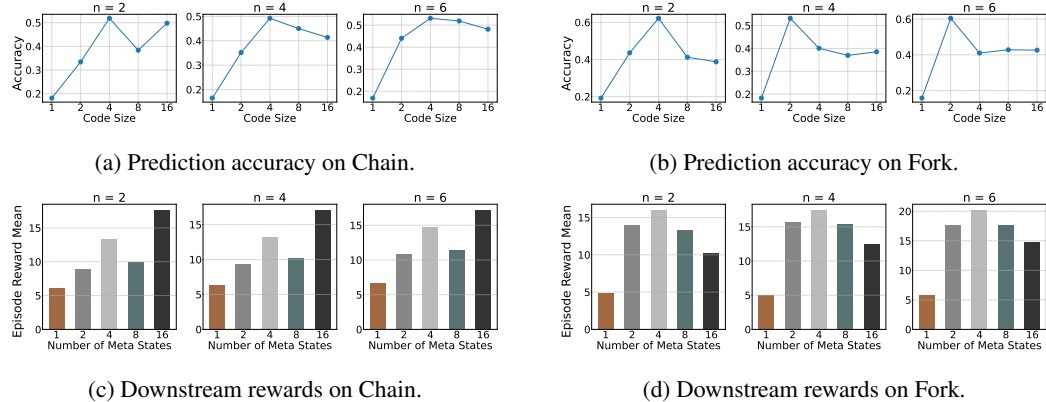

(a) Prediction accuracy on Chain.

(b) Prediction accuracy on Fork.

(c) Downstream rewards on Chain.

(d) Downstream rewards on Fork.

Figure 2: Performance with different numbers of meta states. (a-b) show prediction accuracy on Chain and Fork environments respectively, while (c-d) show corresponding downstream rewards.

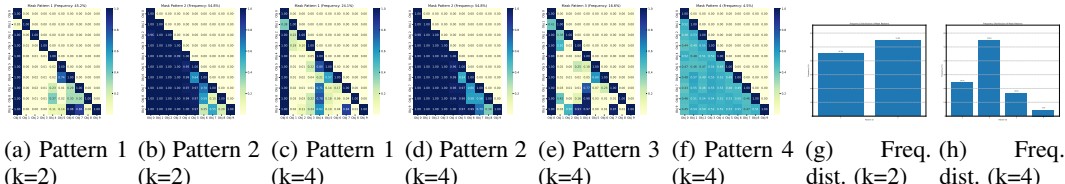

(a) Pattern 1 (k=2)  (b) Pattern 2 (k=2)  (c) Pattern 1 (k=4)  (d) Pattern 2 (k=4)  (e) Pattern 3 (k=4)  (f) Pattern 4 (k=4)  (g) Freq. dist. (k=2)  (h) Freq. dist. (k=4)

Figure 3: Comparison of causal patterns discovered with different numbers of meta states. When the number of meta states is set to 2, the model learns two distinct causal patterns (a,d) that correspond to the meta states. With 4 meta states, the model learns four patterns (b,c,e,f), where some patterns share similarities (Patterns 1 and 3), while others occur with lower frequency (Pattern 4) as shown in the frequency distributions (g,h).

require strong prior knowledge [57, 29] or complete access to a known dynamic model [10], and they lack active, curiosity-driven exploration strategies for open-ended environments [56].

We introduce an intervention-driven framework that empowers an agent to actively probe its surroundings in order to uncover context-dependent causal graphs. By combining targeted interventions with a curiosity-driven exploration policy, our approach adaptively reveals the global causal rules governing rich, previously unseen environments. Extended related work could be found in Appendix C.

## 6 Experiments

We evaluate our method by examining the following research questions: (1) Does our method learn a more accurate causal world model? (Table 1) (2) Does it enhance performance on downstream tasks? (Table 2) (3) How does overparameterization affect our method's performance? (Figure 2) (4) Do the curiosity-driven reward and intervention verification improve the learned causal world model quality? (Table 1, Table 2) (5) Does our method enable more accurate causal subgraph learning? (Figure 5)

### 6.1 Experiment Setup

We compare our method against several baselines: **MLP**, a dense model that predicts transition dynamics $p(x_{t+1}|x_t, a_t)$; **Modular**, a modular network with separate modules for each state variable; **GNN** [40], a graph neural network for predicting transition dynamics; **NPS** [2], which learns sparse and modular dynamics; **CDL** [77], which learns a static causal model from data; **GRADER** [13], which learns a static causal model using conditional independence tests; **Sandy-Mixure** [57], which uses Jacobian matrices of MLP layers to learn the local causal graph; **Transformer** [73], which infers causal graphs by attention; **NCD** [31], a neural causal discovery algorithm for learning causal graphs for each sample; and **FCDL** [31] learns causal graph to facilitate the robustness for reinforcement learning. Please check Appendix B for details of the experimental setup.

**Algorithm 1** Curious Causality-Seeking Meta Causal World Modeling

---

**Require:** Observational states $X$, meta state embedding space $U$, initial codebook embeddings $\{e_u\}$, encoder $E$, decoder $D$, hyperparameters $\lambda_{\text{sparse}}, \lambda_{\text{mask}}, \lambda_{\text{quantization}}$
1: Initialize encoder parameters $\phi$, decoder parameters $\theta$, and embeddings $\{z_u\}_{u \in U}$
2: **while** not converged **do**
3:     Compute embedding assignment: $u \leftarrow \arg\min_{u \in U} \|E_\phi(X) - z_u\|_2^2$
4:     Compute causal skeleton probabilities matrix: $\hat{M}_u \leftarrow D(e_u)$
5:     Sample causal skeleton matrix $M_u[i, j] \sim \text{GumbelSoftmax}(\hat{M}_u[i, j])$
6:     Select intervention $\text{do}(X_i = x_i')$ maximizing reward $R_t$ in Section 4.1
7:     Perform intervention and record resulting state transitions
8:     Estimate causal effects $\Delta_{ij}$ from interventions and update causal mask parameters via $\mathcal{L}_{\text{mask}}$
9:     Predict next state $X_j^{t+1}$ from parent set $Pa_{\mathcal{G}_u}(X_j)$ and update world model via $\mathcal{L}_{\text{MLE}}$
10:    Encourage sparsity in causal graph via $\mathcal{L}_{\text{sparse}}$
11:    Update embedding codebook $\{e_u\}$ via $\mathcal{L}_{\text{quantization}}$
12: **end while**
13: **return** Learned causal subgraphs $\mathcal{G}_u$, world model parameters, and embeddings $\{e_u\}$

---

## 6.2 Environments

**Chemical** [39] We use the Chemical environment to evaluate the performance of the proposed method on learning the causal graphs in a system with multiple causal structures. There are several causal graphs (*full*, *fork*, *chain*) in the Chemical environment and the causal graph depends on the state of the objects. We use two settings of the Chemical environment: (1) *full-fork (Fork)* and (2) *full-chain (Chain)*. **Magnetic** [31] The *Magnetic* environment is a robot arm manipulation task (Figure 4). This environment is built based on the Robosuite suite. Please refer to Appendix B for more details.

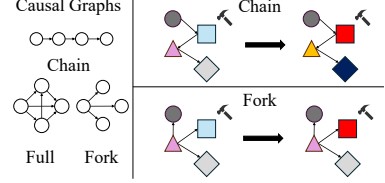

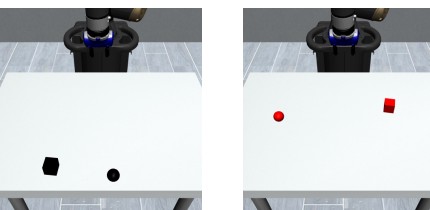

Figure 4: Visualization of environments: (top) **Chemical**; (bottom) **Magnetic**.

## 6.3 Experiment Results

**Prediction Accuracy.** We evaluate the prediction accuracy of our proposed method in comparison to several baseline approaches. To systematically assess robustness, we introduce varying levels of noise to the state information. For the full-fork and full-chain tasks, we corrupt the values of 2, 4, and 6 nodes, respectively, to measure performance degradation under increasing noise. In the magnetic environment, we assess prediction accuracy when box position coordinates and ball/box color properties are corrupted, thereby testing the model's performance under partial observability. All experiments are conducted eight times, with results averaged. As shown in Table 1, our method consistently achieves the highest prediction accuracy across nearly all baselines, demonstrating its effectiveness in learning a more accurate causal world model.

**Downstream Task** We evaluate our method against baselines on the Chemical downstream task, using the trained model to predict future states with the learned model for decision making. As shown in Table 2, our approach achieves the highest reward, demonstrating its effectiveness in learning a more accurate causal world model.

**Overparameterization** We evaluate our method using 2, 4, 8, and 16 meta states, with results presented in Figure 3. Performance declines markedly when using fewer meta states than actually exist, as the model fails to capture distinct causal structures. Conversely, overparameterized models consistently outperform underparameterized ones, empirically validating Theorem 2. The learned causal graphs with codebook sizes of 2 and 4 reveal that with 2 meta states, the model learns two distinct causal graphs (Pattern 1 and Pattern 2) corresponding to the two meta states, consistent

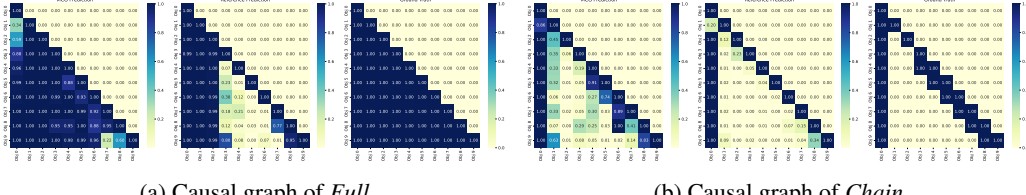

(a) Causal graph of *Full*.          (b) Causal graph of *Chain*.

Figure 5: Comparison of learned causal graphs in the Chemical task: left—MCG (ours); middle—reference (FCDL); right—ground truth.

with Theorem 1. With 4 meta states, some codes correspond to similar underlying causal graphs (as shown by the similarity between Pattern 1 and Pattern 3), while codes representing significantly different causal graphs (such as Pattern 4) occur less frequently, minimally affecting overall model performance. This demonstrates the model's capacity to consolidate redundant representations when overparameterized, aligning with our theoretical findings.

**Ablation Study** We conduct ablation studies to evaluate the effectiveness of the proposed method. We remove the following components from the proposed method: (1) the causal loss $\mathcal{L}_{\mathrm{mask}}$ and (2) the reward function. The first ablation study aims to test the effectiveness of the intervention verification, while the second ablation study aims to test the effectiveness of the active intervention exploration. The results are shown in Table 2. Overall, the proposed method outperforms the ablation methods. The performance drop of removing the causal loss $\mathcal{L}_{\mathrm{mask}}$ indicates that the incorporation of the causal loss can help the model learn a more accurate causal world model. The performance drop of removing the reward function indicates that the curiosity-based reward function can help the agent learn a more accurate causal world model through active exploration.

**Partial Observation** We modify the Chemical environment to explicitly simulate challenges arising from limited observation windows. This experimental setup evaluates how our method performs as new variables gradually become observable. The results indicate that our approach remains robust under partial observability and shows strong potential for handling more realistic, dynamically evolving environments. The results are shown in Table 8.

**Efficiency** We compare baselines with longer training rounds and more parameters to show efficiency of our method. Results are reported in Table 10 and Table 11.

**Causal Subgraph** We visualize the learned causal subgraphs in Figure 5. It is evident that the proposed method successfully captures the underlying causal structures of the environment.

Additional results are in Appendix B and the source code is included in the supplementary materials.

# 7   Conclusion, Limitation and Future Work

In this work, we introduced a method to model the environment using the Meta-Causal Graph that explicitly captures how causal relationships evolve across different environmental contexts-meta state. Our approach addresses two critical limitations in existing causal world models: the inability of uniform causal graphs to capture context-dependent changes and the lack of active exploration making it hard to learn the open-world environment. We theoretically established the identifiability of meta states and their corresponding causal subgraphs, and developed the Curious Causality-Seeking Agent framework that actively explores environments through interventions guided by a curiosity-driven reward function. Our empirical evaluations demonstrated that our method outperforms existing approaches across both synthetic tasks and a challenging robot arm manipulation task.

**Limitation and Future Work** Despite these advances, real-world exploration remains subject to inherent constraints: certain states are fundamentally unreachable via any sequence of interventions, and the implications of such unobservability for causal inference have yet to be analyzed (see Appendix A.8). Moreover, interventions incur costs, and practical agents must operate within limited budgets. Future research will investigate the impact of unreachable states on the identifiability and accuracy of learned causal structures, and will develop budget-aware exploration strategies to optimally allocate limited intervention resources.

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

# A Theoretical Analysis and Discussion

This section establishes the theoretical foundations for identifying meta states and their corresponding causal subgraphs within our proposed framework. We present a comprehensive analysis of identifiability conditions under two distinct scenarios: swap-label equivalence and observational equivalence. The former addresses cases where meta states can be permuted without altering the underlying causal structure, while the latter examines situations involving overparameterization where the learned model contains more meta states than the ground truth. Additionally, we introduce intervention reachability to formalize which causal relationships are identifiable via feasible interventions.

## A.1 Summary of Key Results

Our theoretical analysis yields key results on the identifiability of meta states and causal structures:

**Swap-Label Equivalence:** Under Assumption 1, we demonstrate that the learned mapping function is swap-label equivalent to the ground truth mapping (Theorem 1). This equivalence ensures that while the specific labels assigned to meta states may differ between learned and ground truth mappings, the underlying causal relationships remain preserved. The proof demonstrates that any departure from this equivalence leads to higher L1 norm of causal skeleton matrices and thus contradicts the optimality principle underlying our framework.

**Observational Equivalence:** For overparameterized scenarios where the learned model contains more meta states than the ground truth, we prove that observational equivalence is maintained (Theorem 2). This result demonstrates that introducing additional meta states does not compromise the recovery of true causal relationships, since the learned mapping consistently preserves the same causal skeleton matrices across all states.

**Intervention Reachability:** We formalize the mathematical framework for determining which interventions are feasible given environmental constraints. Through matrix representations of state transitions and intervention capabilities, we establish conditions under which specific causal relationships can be identified through sequential interventions (Theorem 3). This analysis provides practical guidance for experimental design in causal discovery.

**Structural Assumptions:** Our results rely on Assumption 1, which posits that learning from mixed datasets yields causal graphs encompassing the union of all contributing causal relationships. This assumption is both theoretically justified and practically reasonable, as it ensures comprehensive coverage of causal dependencies present in heterogeneous data sources.

The theoretical framework presented in this section provides rigorous guarantees for the identifiability of causal structures while accommodating practical constraints inherent in real-world applications, thereby establishing the soundness of our proposed methodology.

## A.2 Swap-Label Equivalence

In the context of Meta-Causal Graphs, swap-label equivalence captures the invariance of causal structure under permutation of meta state labels. This phenomenon arises when different labelings of meta states yield identical causal relationships across the state space.

Consider a Meta-Causal Graph $G$ with meta states $\{u_1, u_2\}$ and corresponding causal subgraphs $\{M_{u_1}, M_{u_2}\}$, as illustrated in Figure 6a. An alternative Meta-Causal Graph $\hat{G}$ with meta states $\{\hat{u}_1, \hat{u}_2\}$ and subgraphs $\{\hat{M}_{\hat{u}_1}, \hat{M}_{\hat{u}_2}\}$ is shown in Figure 6b.

Two mappings $C : \mathcal{X} \to U$ and $\hat{C} : \mathcal{X} \to \hat{U}$ are swap-label equivalent if there exists a bijection $g : U \to \hat{U}$ such that for any state $x \in \mathcal{X}$, the causal skeleton matrices satisfy:

$$M_{C(x)} = \hat{M}_{g(C(x))} = \hat{M}_{\hat{C}(x)}$$

This equivalence ensures that the learned mapping $\hat{C}$ preserves the same causal structure as the ground truth mapping $C$, despite potentially different meta state assignments. As illustrated in Figure 6, such permutations preserve the fundamental causal relationships within the Meta-Causal Graph.

(a) Two meta states $(u_1, u_2)$ with corresponding subgraphs $M_{u_1}$ and $M_{u_2}$.

(b) Label-swapped meta states $(\hat{u}_1, \hat{u}_2)$ with corresponding subgraphs $\hat{M}_{\hat{u}_1}$ and $\hat{M}_{\hat{u}_2}$.

Figure 6: **Swap-label equivalence in Meta-Causal Graphs.** Left: Ground truth Meta-Causal Graph with meta states $\{u_1, u_2\}$ and corresponding causal subgraphs $\{M_{u_1}, M_{u_2}\}$. Right: Learned Meta-Causal Graph with meta states $\{\hat{u}_1, \hat{u}_2\}$ and subgraphs $\{\hat{M}_{\hat{u}_1}, \hat{M}_{\hat{u}_2}\}$. Despite different meta state labels, both mappings produce identical causal skeleton matrices for any state $x \in \mathcal{X}$, demonstrating structural equivalence under label permutation.

### A.3 Proof of Theorem 1

*Proof.* Consider the reconstruction objective for learning causal skeleton matrices from mixed data. For any clustering $\hat{C} : \mathcal{X} \to \hat{U}$, let $\hat{M}_{\hat{u}}$ denote the learned skeleton matrix for meta state $\hat{u}$. Under Assumption 1, this matrix satisfies:

$$\hat{M}_{\hat{u}}[i, j] = \mathbb{I}\left[\bigcup_{x:\hat{C}(x)=\hat{u}} M_{C(x)}[i, j] = 1\right]$$

Now consider two cases:

**Case 1:** $\hat{C}$ perfectly aligns with $C$ up to label permutation. Then there exists a bijection $g : U \to \hat{U}$ such that $\hat{C}(x) = g(C(x))$ for all $x \in \mathcal{X}$. In this case, $\hat{M}_{g(u)} = M_u$ for each $u \in U$, achieving perfect reconstruction of individual causal structures.

**Case 2:** $\hat{C}$ merges states from different true meta states. Then there exist $x_1, x_2 \in \mathcal{X}$ with $C(x_1) = u_1 \neq u_2 = C(x_2)$ but $\hat{C}(x_1) = \hat{C}(x_2) = \hat{u}$. By Assumption 1:

$$\hat{M}_{\hat{u}} = M_{u_1} \cup M_{u_2}$$

where $\cup$ denotes element-wise logical OR. Since $M_{u_1} \neq M_{u_2}$, we have $\hat{M}_{\hat{u}} \neq M_{u_1}$ and $\hat{M}_{\hat{u}} \neq M_{u_2}$.

Let $\|\cdot\|_1$ denote the L1 norm. For binary matrices, $\|M\|_1$ equals the number of edges in the corresponding causal graph. Since the union operation can only add edges (never remove them), we have:

$$\|\hat{M}_{\hat{u}}\|_1 = \|M_{u_1} \cup M_{u_2}\|_1 \geq \max\{\|M_{u_1}\|_1, \|M_{u_2}\|_1\}$$

Moreover, since $M_{u_1} \neq M_{u_2}$, there exists at least one position $(i, j)$ where $M_{u_1}[i, j] \neq M_{u_2}[i, j]$.

$$\|\hat{M}_{\hat{u}}\|_1 > \|M_{u_1}\|_1 \quad \text{and} \quad \|\hat{M}_{\hat{u}}\|_1 > \|M_{u_2}\|_1$$

Consider the expected L1 norm under each partition. For the true partition $C$:

$$\mathbb{E}_{x \sim \mathcal{X}}[\|M_{C(x)}\|_1] = \sum_{u \in U} P(C(x) = u) \cdot \|M_u\|_1$$

For the suboptimal partition $\hat{C}$ that merges distinct meta states:

$$\mathbb{E}_{x \sim \mathcal{X}}[\|\hat{M}_{\hat{C}(x)}\|_1] = \sum_{\hat{u} \in \hat{U}} P(\hat{C}(x) = \hat{u}) \cdot \|\hat{M}_{\hat{u}}\|_1$$

Since merging increases the L1 norm (as shown above), and each merged cluster $\hat{u}$ has $\|\hat{M}_{\hat{u}}\|_1 > \|M_u\|_1$ for the constituent true meta states $u$, we have:

$$\mathbb{E}_{x \sim \mathcal{X}}[\|\hat{M}_{\hat{C}(x)}\|_1] > \mathbb{E}_{x \sim \mathcal{X}}[\|M_{C(x)}\|_1]$$

$\square$

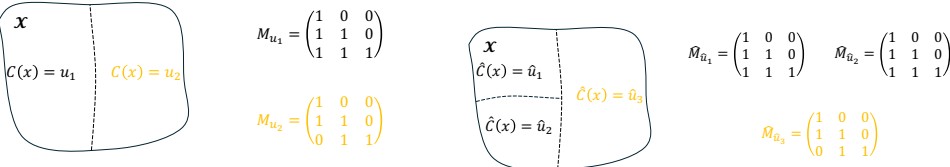

(a) Two meta states: partition (left) and Meta-Causal Graph (right).

(b) Three meta states: partition (left) and Meta-Causal Graph (right).

Figure 7: **Observational equivalence under overparameterization.** Left: Ground truth Meta-Causal Graph with meta states $\{u_1, u_2\}$ and causal subgraphs $\{M_{u_1}, M_{u_2}\}$. Right: Overparameterized Meta-Causal Graph with three meta states $\{\hat{u}_1, \hat{u}_2, \hat{u}_3\}$ and subgraphs $\{\hat{M}_{\hat{u}_1}, \hat{M}_{\hat{u}_2}, \hat{M}_{\hat{u}_3}\}$. Despite the additional meta state, both mappings produce identical causal skeleton matrices for any state $x \in \mathcal{X}$, demonstrating that overparameterization preserves causal structure.

## A.4 Observational Equivalence

Observational equivalence captures scenarios where learned meta state mappings preserve causal structure despite overparameterization. Formally, two mappings $C : \mathcal{X} \to U$ and $\hat{C} : \mathcal{X} \to \hat{U}$ are observationally equivalent if for every state $x \in \mathcal{X}$: $M_{C(x)} = \hat{M}_{\hat{C}(x)}$, where $M_{C(x)}$ and $\hat{M}_{\hat{C}(x)}$ are the causal skeleton matrices corresponding to the assigned meta states. This equivalence ensures that the learned mapping $\hat{C}$ captures identical causal relationships as the ground truth mapping $C$, even when the learned meta state space $\hat{U}$ is larger than the true space $U$. Overparameterization may result in redundant meta states, but the essential causal structure remains preserved.

Figure 7 illustrates this concept. Left: Ground truth Meta-Causal Graph with meta states $\{u_1, u_2\}$ and subgraphs $\{M_{u_1}, M_{u_2}\}$. Right: Learned overparameterized Meta-Causal Graph with three meta states $\{\hat{u}_1, \hat{u}_2, \hat{u}_3\}$ and corresponding subgraphs $\{\hat{M}_{\hat{u}_1}, \hat{M}_{\hat{u}_2}, \hat{M}_{\hat{u}_3}\}$. Despite having an additional meta state, both mappings yield identical causal skeleton matrices for any given state $x \in \mathcal{X}$, demonstrating observational equivalence under overparameterization.

## A.5 Proof of Theorem 2

*Proof.* We prove that despite overparameterization, the learned mapping preserves the causal structure of the ground truth mapping.

Let $\hat{U}_{\text{active}} = \{\hat{u} \in \hat{U} : \exists x \in \mathcal{X} \text{ s.t. } \hat{C}(x) = \hat{u}\}$ denote the set of meta states that are actually assigned to some state in $\mathcal{X}$.

**Step 1:** We first establish that the learned mapping cannot merge distinct true meta states. Suppose for contradiction that there exist $x_1, x_2 \in \mathcal{X}$ such that:

$$C(x_1) = u_1 \neq u_2 = C(x_2) \quad \text{but} \quad \hat{C}(x_1) = \hat{C}(x_2) = \hat{u}$$

By Assumption 1, the learned causal skeleton matrix would be:

$$\hat{M}_{\hat{u}} = M_{u_1} \cup M_{u_2}$$

Since $M_{u_1} \neq M_{u_2}$, we have $\|\hat{M}_{\hat{u}}\|_1 > \max\{\|M_{u_1}\|_1, \|M_{u_2}\|_1\}$, increasing the expected structural complexity as shown in Theorem 1. This contradicts optimal clustering.

**Step 2:** Since distinct true meta states cannot be merged, each active learned meta state $\hat{u} \in \hat{U}_{\text{active}}$ corresponds to exactly one true meta state. That is, for each $\hat{u} \in \hat{U}_{\text{active}}$, there exists a unique $u \in U$ such that: $\{x : \hat{C}(x) = \hat{u}\} \subseteq \{x : C(x) = u\}$

**Step 3:** For any $x \in \mathcal{X}$, if $\hat{C}(x) = \hat{u}$ and the corresponding true meta state is $u$, then by Assumption 1:

$$\hat{M}_{\hat{u}} = M_u$$

Therefore, $M_{C(x)} = \hat{M}_{\hat{C}(x)}$ for all $x \in \mathcal{X}$, establishing observational equivalence.

(a) Ground truth: graph (left) and skeleton matrix (right).

(b) Over-parameterized: learned graph (left) and skeleton matrix (right).

Figure 8: Illustration of Theorem 2. Each pair places the graph (left) alongside its causal skeleton matrix (right): *(a)* ground truth; *(b)* overparameterized solution.

$\square$

## A.6 Disscussion on Assumption 1

**Assumption 2.** *Let $\mathcal{MG} = \{\mathcal{G}_i\}_K$ be a Meta-Causal Graph with $K$ causal subgraphs corresponding to distinct meta states $u \in U$. Consider a dataset $\mathcal{D}$ where each sample is generated from one of these causal subgraphs. Let $S_\mathcal{D} \subseteq \{1, 2, ..., K\}$ denote the indices of causal subgraphs that actually contributed samples to $\mathcal{D}$. When learning a single causal graph $\hat{\mathcal{G}}$ from the mixed dataset $\mathcal{D}$ (treating all samples as if they were generated from a single causal subgraph), the estimated parent set for each variable $X_j$ in $\hat{\mathcal{G}}$ satisfies:*

$$Pa_{\hat{\mathcal{G}}}(X_j) = \bigcup_{i \in S_\mathcal{D}} Pa_{\mathcal{G}_i}(X_j) \quad \forall j \in [p]. \tag{2}$$

Assumption 1 states that learning from mixed data generated by multiple causal subgraphs yields a unified graph capturing the union of all parent-child relationships across contributing subgraphs.

Figure 10 illustrates this assumption. The left panel shows the causal skeleton matrices $M_u$ of individual subgraphs that contribute data to $\mathcal{D}$. The right panel shows the learned causal skeleton matrix $\hat{M}$ from the pooled dataset. Red elements highlight the union property where edges present in any contributing subgraph appear in the learned graph.

Mathematically, this relationship can be expressed as:

$$\hat{M}[i, j] = \mathbb{I}\left[\bigcup_{u \in S_\mathcal{D}} M_u[i, j] = 1\right] \quad \forall i, j \in [p], \tag{3}$$

where $\mathbb{I}[\cdot]$ is the indicator function, ensuring that an edge $(i, j)$ exists in the learned graph if and only if it exists in at least one contributing subgraph.

This assumption is well-motivated: when data from multiple causal mechanisms are pooled without knowledge of their source, standard causal discovery algorithms tend to include all statistically supported edges to avoid missing true causal relationships. Violating this assumption would imply that the learning algorithm systematically ignores genuine causal relationships present in the data, leading to incomplete and potentially misleading causal models.

Assumption 1 shows that the L1 norm of the learned causal skeleton matrix is larger than the L1 norm of the causal skeleton matrix of the contributing graphs. This also provides a way to show that the underparameterization of meta states will lead to the failure of learning the causal graph.

Figure 9 shows an example of underparameterization. The upper figure shows the ground truth causal graph with three meta states and the corresponding causal skeleton matrix. The lower figure shows the learned graph with two meta states and the corresponding causal skeleton matrix. The underparameterization of the learned graph leads to the learned causal skeleton matrix different from the ground truth causal skeleton matrix (Equation 3).

However, overparameterization does not lead to the failure of learning the causal graph. Figure 8 shows an example of overparameterization. Although we may assign different meta states to states which are generated from the same causal graph, the data to learn each causal graph is still generated from the same causal graph.

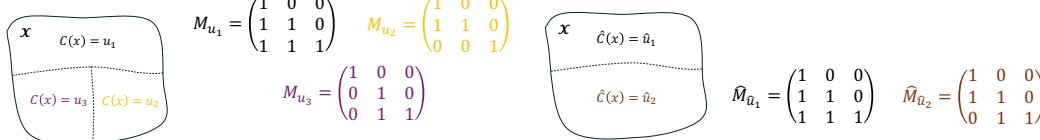

(a) Three meta states: causal graph (left) and learned graph (right).

(b) Two meta states: causal graph (left) and learned graph (right).

Figure 9: Illustration of the Assumption 1. The left figure shows the ground truth causal graph with three meta states and the corresponding causal skeleton matrix. The right figure shows the learned graph with two meta states and the corresponding causal skeleton matrix.

## A.7 Proof of Proposition 1

*Proof.* We prove that the given intervention condition ensures unique identifiability by showing that no two distinct DAGs can be interventionally Markov equivalent under $\mathcal{I}$.

Let $\mathcal{D}_{\mathcal{I}}$ denote the interventional Markov equivalence class (I-MEC) of DAGs that are interventionally Markov equivalent to $D$ given $\mathcal{I}$.

**Step 1:** Consider any edge $i \to j$ in the true causal graph $D$. By assumption, there exists $I \in \mathcal{I}$ such that exactly one of $\{i, j\}$ is in $I$.

**Case 1:** If $i \in I$ and $j \notin I$, then in the intervention graph $D^{(I)}$, all edges into node $i$ are removed, but the edge $i \to j$ (if it exists) remains. Consider any DAG $D'$ with the reversed edge $j \to i$. In $D'^{(I)}$, this edge would be removed since $i \in I$, creating different intervention graphs: $D^{(I)} \neq D'^{(I)}$.

$$M_{u_1} = \begin{pmatrix} 1 & 0 & 0 \\ 1 & 1 & 0 \\ 0 & 0 & 1 \end{pmatrix}$$

$$M_{u_2} = \begin{pmatrix} 1 & 0 & 0 \\ 0 & 1 & 0 \\ 0 & 1 & 1 \end{pmatrix}$$

$$\widehat{M}_u = \begin{pmatrix} 1 & 0 & 0 \\ 1 & 1 & 0 \\ 0 & 1 & 1 \end{pmatrix}$$

Figure 10: Illustration of the Assumption 1. The left side shows the causal skeleton matrices of causal graphs which contribute to the dataset for generating the data. The right side shows the learned causal skeleton matrices of the causal graph from the dataset. The elements in red highlight the differences between them.

**Case 2:** If $j \in I$ and $i \notin I$, then in $D^{(I)}$, all edges into $j$ (including $i \to j$) are removed. A DAG $D'$ with edge $j \to i$ would retain this edge in $D'^{(I)}$ since $i \notin I$, again yielding $D^{(I)} \neq D'^{(I)}$.

**Step 2:** Since every edge $i \to j \in D$ satisfies the intervention condition, every edge direction is uniquely determined by the interventional data, ensuring that for any DAG $D' \in \mathcal{D}_{\mathcal{I}}$, we have $D^{(I)} = D'^{(I)}$ for all $I \in \mathcal{I}$.

**Step 3:** The intervention condition ensures that any edge orientation different from that in $D$ would create distinguishable intervention graphs, contradicting interventional Markov equivalence. Therefore, $|\mathcal{D}_{\mathcal{I}}| = 1$, meaning $D$ is uniquely identifiable. □

**Remark 1.** *Score-based causal discovery methods identify structures only up to a Markov equivalence class, since different directed graphs can encode the same set of conditional independencies. Interventions, however, can break this equivalence: by actively perturbing variables, one can distinguish among graphs within the same class. For instance, in a causal chain $A \to B \to C$, an intervention on $A$ affects $C$ only through $B$. If $B$ is not held fixed, a purely observational method might incorrectly infer a direct edge $A \to C$. To fully recover the underlying structure, multiple interventions, potentially on more than one variable, are required.*

*In our framework, this issue is addressed in two complementary steps. First, using interventional comparison (Proposition 1), we detect edge existence by contrasting causal subgraphs obtained from distinct interventions. Second, within each intervention subgraph, we apply a score-based refinement guided by structural minimality (Theorem 1). This ensures that adding spurious edges*

*such as $A \to C$ would increase model complexity without improving fit, allowing the true causal structure to be uniquely identified.*

## A.8  Reachability of Interventions

Environmental constraints often prevent direct manipulation of every state variable in practice. We consider a subset $\mathcal{S}_c \subseteq [p]$ of variables permitting direct intervention, while others require multi-step sequences or remain unreachable. We define *intervention reachability*: a state is reachable if attainable from the current state via finite allowable intervention sequences.

Each state variable $X_i$ takes $n_i$ discrete values, represented using one-hot encoding as $\mathbf{z}_i \in \{0, 1\}^{n_i}$ where exactly one element equals 1. The complete system state is $\mathbf{z} = \mathbf{z}_1 \otimes \mathbf{z}_2 \otimes \cdots \otimes \mathbf{z}_p \in \{0, 1\}^N$, where $\otimes$ denotes the Kronecker product and $N = \prod_{i=1}^{p} n_i$.

**Remark 2.** *The Kronecker product of one-hot vectors remains one-hot. If $\mathbf{z}_1 \in \{0, 1\}^m$ and $\mathbf{z}_2 \in \{0, 1\}^n$ are one-hot vectors, then $\mathbf{z} = \mathbf{z}_1 \otimes \mathbf{z}_2 \in \{0, 1\}^{mn}$ is also one-hot, with $\mathbf{z}[nr+v] = \mathbf{z}_1[r]\mathbf{z}_2[v]$ for $r \in [m], v \in [n]$. Since $\mathbf{z}_1$ and $\mathbf{z}_2$ are one-hot encoded vectors, we have $\mathbf{z}_1[r] \in \{0, 1\}$ and $\mathbf{z}_2[v] \in \{0, 1\}$.*

*Therefore, $\mathbf{z}[nr + v] \in \{0, 1\}$ and there exists only one $nr + v$ such that $\mathbf{z}[nr + v] = 1$.*

*Thus, $\mathbf{z}$ is a one-hot encoded vector. Therefore, the Kronecker product of one-hot encoded vectors is still a one-hot encoded vector.*

*The Kronecker product $\mathbf{z}$ is a one-hot encoded vector, which can represent the state of the system.*

We define two key matrices:

- **Intervention Matrix** $F \in \{0, 1\}^{N \times N}$: $F[i, j] = 1$ if state $j$ can be directly reached from state $i$ through intervention on variables in $\mathcal{S}_c$.
- **Transition Matrix** $T \in \{0, 1\}^{N \times N}$: $T[i, j] = 1$ if the system naturally transitions from intervened state $i$ to state $j$.

Given a state vector $\mathbf{z} \in \{0, 1\}^N$, the intervention operation $\mathbf{z}' = F\mathbf{z}$ produces:

$$\mathbf{z}'[k] = \sum_{i=1}^{N} F[k, i]\mathbf{z}[i]$$

Since $\mathbf{z}$ is one-hot, $\mathbf{z}'[k] > 0$ if and only if there exists an index $i$ such that $\mathbf{z}[i] = 1$ and $F[k, i] = 1$. This means state $k$ is directly reachable from the current state $i$ through intervention. The non-zero elements of $F\mathbf{z}$ thus indicate all states achievable by a single intervention step.

Similarly, the transition operation $\mathbf{z}'' = T\mathbf{z}'$ yields:

$$\mathbf{z}''[k] = \sum_{i=1}^{N} T[k, i]\mathbf{z}'[i]$$

Here, $\mathbf{z}''[k] > 0$ indicates that state $k$ can be reached from some intervened state $i$ where $\mathbf{z}'[i] > 0$ and $T[k, i] = 1$. This captures the natural system dynamics following intervention.

Combining these operations, the complete system evolution under intervention is:

$$\mathbf{z}_{t+1} = TF\mathbf{z}_t$$

This composition first applies interventions (via $F$) to determine immediately accessible states, then applies system dynamics (via $T$) to find the resulting states after natural transitions. The non-zero elements in $(TF)^k \mathbf{z}_0$ represent all states reachable within $k$ intervention-transition cycles from $\mathbf{z}_0$.

**Theorem 3** (Reachability Analysis). *A state corresponding to index $i$ is reachable from initial state $\mathbf{z}_0$ if and only if there exists $k \geq 0$ such that $[(TF)^k \mathbf{z}_0][i] > 0$.*

*Proof.* We prove both directions of the equivalence.

(⇒): If state $i$ is reachable from $\mathbf{z}_0$, then by definition there exists a finite sequence of intervention-transition steps that leads from $\mathbf{z}_0$ to a state where the $i$-th component is active. This sequence corresponds to some power $k$ of the operator $TF$, hence $[(TF)^k \mathbf{z}_0][i] > 0$.

(⇐): If $[(TF)^k \mathbf{z}_0][i] > 0$ for some $k \geq 0$, then by the definition of matrix-vector multiplication, there exists a computational path through $k$ iterations of intervention-transition operations that activates the $i$-th state component, demonstrating reachability. □

The reachability analysis has direct consequences for causal structure learning:

**Corollary 1** (Intervention Feasibility). *The state corresponding to index $i$ is feasible from initial state $\mathbf{z}_0$ by intervention if and only if there exists $k \geq 0$ such that $[F(TF)^k \mathbf{z}_0][i] > 0$.*

*Proof.* This follows directly from Theorem 3. Here, $F(TF)^k \mathbf{z}_0$ represents states accessible for intervention after $k$ cycles, where the leading $F$ captures the final intervention step. □

This constraint fundamentally limits causal discovery scope in practical settings. The Curious Causality-Seeking Agent must therefore operate within feasible intervention constraints while maximizing causal structure identification. We provide an example to illustrate these results.

**Case 1.** *Consider a system with two binary variables $x_1, x_2$, yielding four possible states as shown in Table 3. As $z_1$ and $z_2$ are one-hot encoded vectors, the Kronecker product of $z_1$ and $z_2$ is a*

Table 3: One-hot encoding of $x_1$ and $x_2$ and their Kronecker product.

| $x_1$ | $x_2$ | $z_1$ | $z_2$ | $z = z_1 \otimes z_2$ |
|---|---|---|---|---|
| 0 | 0 | $[1,0]^\top$ | $[1,0]^\top$ | $[1,0,0,0]^\top$ |
| 1 | 0 | $[0,1]^\top$ | $[1,0]^\top$ | $[0,1,0,0]^\top$ |
| 0 | 1 | $[1,0]^\top$ | $[0,1]^\top$ | $[0,0,1,0]^\top$ |
| 1 | 1 | $[0,1]^\top$ | $[0,1]^\top$ | $[0,0,0,1]^\top$ |

*4-dimensional one-hot encoded vector, which can represent the state of the system. The space of the system can be represented as $\mathcal{Z} = \{[1,0,0,0]^\top, [0,1,0,0]^\top, [0,0,1,0]^\top, [0,0,0,1]^\top\}$. For $z_i, z_j \in \mathcal{Z}$, if $z_j$ can be reached from $z_i$ by intervening on $x_i$, we can denote this as $F[i,j] = 1$.*

***Intervention Constraints:*** *Suppose only $x_1$ can be directly intervened. The intervention matrix $F$ allows transitions between states that differ only in $x_1$:*

$$F = \begin{bmatrix} 1 & 1 & 0 & 0 \\ 1 & 1 & 0 & 0 \\ 0 & 0 & 1 & 1 \\ 0 & 0 & 1 & 1 \end{bmatrix}$$

***System Dynamics:*** *After intervention, assume the system exchanges the values of $x_1$ and $x_2$. The transition matrix becomes:*

$$T = \begin{bmatrix} 1 & 0 & 0 & 0 \\ 0 & 0 & 1 & 0 \\ 0 & 1 & 0 & 0 \\ 0 & 0 & 0 & 1 \end{bmatrix}$$

***Reachability Analysis:*** *Starting from $\mathbf{z}_0 = [1,0,0,0]^\top$ (state $(0,0)$):*

*Step 1: Direct intervention possibilities:*

$$F\mathbf{z}_0 = \begin{bmatrix} 1 \\ 1 \\ 0 \\ 0 \end{bmatrix}$$

*States 1 and 2 are accessible for intervention.*

*Step 2: After one intervention-transition cycle:*

$$TF\mathbf{z}_0 = \begin{bmatrix} 1 \\ 0 \\ 1 \\ 0 \end{bmatrix}$$

*The system can reach states 1 and 3.*

*Step 3: Interventions possible from these new states:*

$$F(TF\mathbf{z}_0) = F \begin{bmatrix} 1 \\ 0 \\ 1 \\ 0 \end{bmatrix} = \begin{bmatrix} 1 \\ 1 \\ 1 \\ 1 \end{bmatrix}$$

*All states become reachable for intervention within two cycles, demonstrating that despite initial constraints, the system's dynamics enable full state space exploration.*

## A.9    Action as Intervention

Our method identifies context-dependent causal graphs (meta-graphs) via interventions. Actions are how the curiosity-driven agent realizes those interventions. When actions can directly intervene each state (e.g., a chemical environment), we deploy targeted interventions to isolate edges and quickly distinguish meta states and their subgraphs. When not all interventions are feasible, we rely on the reachability assumption.

**Verification Results.**    We add a new experiment based on a manipulation task to demonstrate that our method can handle cases where actions cannot intervene on all variables. We also modify this environment to test generalization. The results show that our method maintains strong performance and generalizes well, even when only a subset of variables is directly intervenable.

Table 4: Performance on partial-intervention manipulation tasks.

| Reward | Small Magnetic Force ($\times 0.02$) | High Ball Density ($\times 10$) | Extra Table Friction |
|---|---|---|---|
| MLP | $4.011 \pm 0.030$ | $3.999 \pm 0.038$ | $3.886 \pm 0.241$ |
| FCDL | $4.482 \pm 0.622$ | $4.451 \pm 0.657$ | $4.193 \pm 0.655$ |
| MCG | $\mathbf{6.173 \pm 0.255}$ | $\mathbf{5.083 \pm 0.245}$ | $\mathbf{5.517 \pm 0.134}$ |

## A.10    Lossy Representation and Misclassification Probability

We analyze how lossy representation affects the probability of mapping samples to prototype embeddings. Table 5 summarizes the key notations used in this derivation.

We assume that for each state $x$ with true meta state $C(x) = u$, the probability that it is mapped by the encoder to the codebook entry index $\hat{u}_k$ is $p_k^u = P(\hat{C}(x) = \hat{u}_k \mid C(x) = u)$. This probability reflects the representation power and discriminability of the encoder, indicating how likely samples of the same true meta state are mapped to different codebook entry.

From Definition 5, $\hat{C}(x)$ and $C(x)$ are *observationally equivalent* if, for all $\hat{u}_k \in \hat{U}$ and all $x_1, x_2 \in \{x \mid \hat{C}(x) = \hat{u}_k\}$, it holds that $C(x_1) = C(x_2)$.

The probability that $\hat{C}(x)$ and $C(x)$ are observationally equivalent is given by

$$P(C(x) = u, \forall x' \in x \mid \hat{C}(x) = \hat{u}_k, C(x') = u) = (1 - p_k + p_k^u \mu_u)^n,$$

where $n$ is the number of samples.

Hence, the overall probability of observational equivalence is

$$P_{\text{equiv}} = \sum_k p_k \sum_{u \in U} \mu_u (1 - p_k + p_k^u \mu_u)^n.$$

Table 5: Notation summary.

| Symbol | Meaning |
|---|---|
| $U = u$ | Ground-truth meta states |
| $\hat{U} = \hat{u}_k$ | Prototype (codebook) embeddings |
| $C(x)$ | True meta state of sample $x$ |
| $\hat{C}(x)$ | Codebook entry selected by encoder |
| $\mu_u$ | Prior probability of state $x$ with true meta state $u$ |
| $p_k = \sum_{u \in U} \mu_u p_k^u$ | Probability of mapping sample $x$ to codebook entry $e_{\hat{u}_k}$ |

The probability that all samples assigned to the same codebook entry index $\hat{C}(x)$ belong to different true meta states is

$$P_{\text{diff}} = \sum_k \sum_{u \in U} (p_k - \mu_u p_k^u)(1 - p_k + p_k^u \mu_u)^n.$$

The probability of misclassification, i.e., that a codebook entry corresponds to mixed true meta states, is

$$P_{\text{misclass}} = 1 - \sum_k p_k \sum_{u \in U} (1 - p_k + p_k^u \mu_u)^n.$$

Considering lossy representation, the misclassification probability simplifies to

$$P(\text{misclassification}) = 1 - \sum_k \sum_{u \in U} \mu_u p_k^u (1 - p_k + p_k^u \mu_u)^n.$$

If $n(p_k + p_k^u \mu_u) \ll 1$, we approximate

$$P(\text{misclassification}) \approx n \sum_u \sum_k \mu_u p_k^u (p_k - p_k^u \mu_u) = \sum_k \left[ p_k^2 - \sum_u (\mu_u p_k^u)^2 \right].$$

In a more accurate representation, for each codebook entry index $k$, the distribution of $\mu_u p_k^u$ concentrates on a single true state—one $p_k^u$ increases while others decrease, thereby increasing $\sum_u (\mu_u p_k^u)^2$ and reducing each term $\left[ p_k^2 - \sum_u (\mu_u p_k^u)^2 \right]$. Consequently, higher representation accuracy monotonically decreases the overall misclassification probability, reaching zero when each prototype corresponds purely to one true meta state.

## B Experimental Details

### B.1 Environment

**Chemical.** We evaluate our method's performance on learning context-dependent causal structures using the Chemical environment [39]. This environment consists of 10 objects, each capable of taking one of 5 color states. An action selects a target object and changes its state, triggering cascading changes to all dependent objects according to the underlying causal graph. The objective is to match each object's color to a specified target configuration.

The Chemical environment features multiple causal structures that switch dynamically based on the system state, making it ideal for evaluating Meta-Causal Graph learning. We consider two experimental settings in this environment:

**Full-Fork (Fork):** The causal structure alternates between two graphs depending on the root node's color. When the root node is red, a fork structure is active; otherwise, a fully-connected structure governs the system dynamics.

**Full-Chain (Chain):** The causal structure alternates between full and chain configurations. A red root node activates the chain structure, while other colors trigger the fully-connected structure.

These settings test our agent's ability to: (1) discover the latent meta states (root node colors) that determine causal structure transitions, (2) learn the corresponding causal subgraphs through

interventional exploration, and (3) generalize to unseen state configurations during evaluation. During the test, some nodes are corrupted with noise. The agent needs to learn the causal graph to match the colors of nodes to the target. The agent starts from a random color configuration and must transform a 10-node graph to match a target color pattern. Actions intervene on nodes, changing their color and that of all their descendants according to the hidden causal graph. The episode reward is the negative Hamming distance to the target (0 for a perfect match).

**Magnetic.** The *Magnetic* environment is a robot arm manipulation task built on the Robosuite framework (Figure 4). The environment contains two objects: a fixed box and a movable ball, whose colors indicate their magnetic properties. When both objects are red, they exhibit magnetic attraction, causing the ball's trajectory to be influenced by the box's position. The magnetic properties of each object are randomly assigned at the beginning of each episode.

## B.2 Reinforcement Learning Algorithm

For fair comparison, we use the same model based reinforcement learning algorithm for all the baselines and our method. We use the cross-entropy method (CEM) [61] to sample the action based on the predicted transition dynamics. The detailed hyper-parameters are shown in the Table 6.

Table 6: CEM parameter.

| CEM parameters | Chemical | | Magnetic |
| --- | --- | --- | --- |
| | full-fork | full-chain | |
| Planning length | 3 | 3 | 1 |
| Number of candidates | 64 | 64 | 64 |
| Number of top candidates | 32 | 32 | 32 |
| Number of iterations | 5 | 5 | 5 |
| Exploration noise | N/A | N/A | 1e-4 |
| Exploration probability | 0.05 | 0.05 | N/A |
| Action type | Discrete | Discrete | Continous |

## B.3 Environment Configurations

The detailed environment configurations are shown in Table 7.

Table 7: Environment configurations

| Paramters | Chemical | | Magnetic |
| --- | --- | --- | --- |
| | Fork | Chain | |
| Training step | $1.5 \times 10^5$ | $1.5 \times 10^5$ | $2 \times 10^5$ |
| Optimizer | Adam | Adam | Adam |
| Learning rate | 1e-4 | 1e-4 | 1e-4 |
| Batch size | 256 | 256 | 256 |
| Initial step | 1000 | 1000 | 1500 |
| Max episode length | 25 | 25 | 25 |
| Action type | Discrete | Discrete | Continous |

## B.4 Partial Observability

Table 8: Prediction accuracy under partial observability in the Chemical environment.

| Prediction Accuracy | $n = 2$ | $n = 4$ | $n = 6$ |
| --- | --- | --- | --- |
| MLP | $31.93 \pm 0.16$ | $32.47 \pm 1.61$ | $33.72 \pm 2.11$ |
| FCDL | $63.75 \pm 13.75$ | $53.89 \pm 11.95$ | $52.11 \pm 16.43$ |
| **MCG (ours)** | **73.52 ± 9.56** | **63.71 ± 5.79** | **57.05 ± 14.63** |

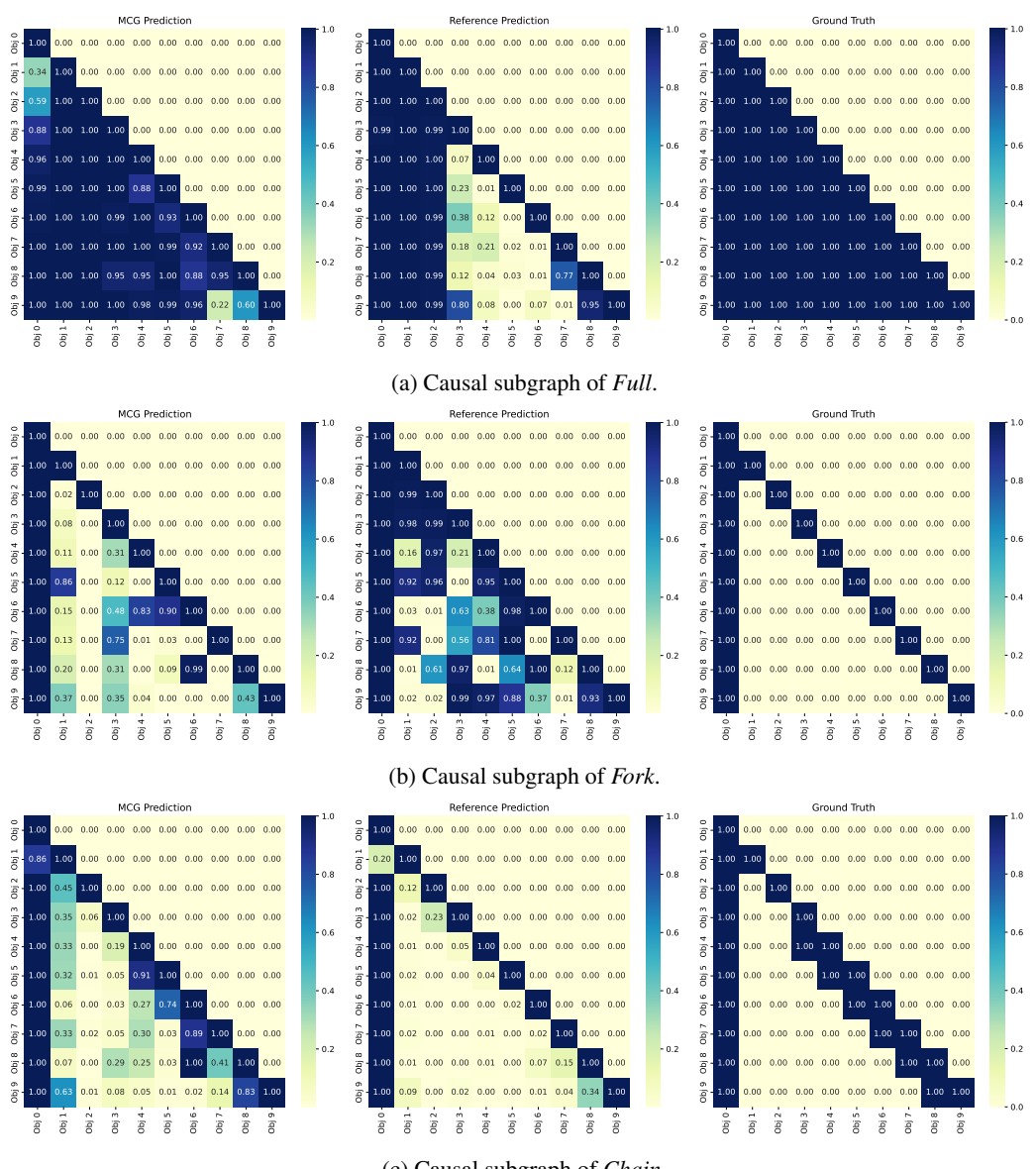

(a) Causal subgraph of *Full*.

(b) Causal subgraph of *Fork*.

(c) Causal subgraph of *Chain*.

Figure 11: Comparison of learned causal graphs in task Chemical. MCG (ours) is the proposed method and the reference is the causal graph learned by FCDL.

## B.5 Extended Results

We visualize the learned causal subgraphs in Figure 11. The results demonstrate that our method effectively captures the underlying causal structures of the environment.

## B.6 Compute Resources and Environment Details

Most experiments were conducted on a server equipped with an AMD EPYC 7V13 64-Core Processor (24 physical cores), supporting 32-bit and 64-bit modes, with 96 MiB L3 cache and 12 MiB L2 cache. The machine was equipped with an NVIDIA A100 PCIe GPU with 80GB memory (driver version 575.51.03, CUDA version 12.9).

## B.7 Code and Demo

The code is available at `https://github.com/zhiyu-zhao-ucas/Meta-Causal-Graph.git`, and demonstrations can be found at `https://sites.google.com/view/meta-causal-world`.

## B.8 Prediction accuracy for three intrinsic terms

Table 9: Prediction accuracy for three intrinsic terms. All values are reported in percentage %.

| Intrinsic Term | Fork | | | Chain | | |
|---|---|---|---|---|---|---|
| | $n=2$ | $n=4$ | $n=6$ | $n=2$ | $n=4$ | $n=6$ |
| Prediction Uncertainty | $52.25 \pm 16.73$ | $59.46 \pm 22.38$ | $52.25 \pm 28.46$ | $59.52 \pm 8.82$ | $54.81 \pm 5.47$ | $59.28 \pm 5.03$ |
| Feature Discrepancy | $46.11 \pm 8.38$ | $41.17 \pm 5.67$ | $43.67 \pm 2.47$ | $56.01 \pm 13.13$ | $51.88 \pm 7.22$ | $58.27 \pm 8.53$ |
| Predictive-Distribution Discrepancy | $54.23 \pm 8.72$ | $46.27 \pm 9.74$ | $43.92 \pm 1.85$ | $36.83 \pm 3.11$ | $40.33 \pm 4.89$ | $45.40 \pm 2.83$ |
| Edge-Entropy | $63.18 \pm 13.94$ | $50.47 \pm 9.87$ | $50.04 \pm 8.56$ | $51.99 \pm 6.58$ | $49.78 \pm 4.11$ | $49.69 \pm 5.14$ |

## B.9 Efficiency

We compare other methods (GNN and MLP) with longer training rounds and more parameters to show efficiency of our method. Results are reported in Table 10 and Table 11.

Table 10: Comparison under the same model size. We evaluate the efficiency of our method by comparing MCG with GNN and MLP trained for longer rounds and with larger parameter counts.

| | Acc n=2 | Acc n=4 | Acc n=6 | Reward n=2 | Reward n=4 | Reward n=6 |
|---|---|---|---|---|---|---|
| GNN (30k round) | 38.13 | 25.81 | 20.75 | 6.83 | 6.06 | 6.83 |
| GNN (25k round) | 36.37 | 26.55 | 20.73 | 6.84 | 5.98 | 6.19 |
| MLP (30k round) | 30.47 | 28.94 | 27.84 | 6.20 | 6.13 | 6.71 |
| MLP (25k round) | 31.47 | 30.32 | 30.42 | 6.18 | 6.03 | 6.23 |
| **MCG (10k round)** | **47.01** | **48.37** | **46.35** | **11.96** | **10.83** | **10.70** |
| **MCG (12k round)** | **48.30** | **48.40** | **47.28** | **12.45** | **10.77** | **10.84** |
| **MCG (15k round)** | **71.56** | **59.17** | **49.58** | **14.65** | **14.06** | **13.28** |

Table 11: Comparison under the same number of training rounds. MCG is compared with GNN and MLP models trained for the same number of rounds but with different model sizes (parameter counts shown in parentheses).

| | Acc n=2 | Acc n=4 | Acc n=6 | Reward n=2 | Reward n=4 | Reward n=6 |
|---|---|---|---|---|---|---|
| MLP (parameter $\times 3.71$) | 24.65 | 25.05 | 23.71 | 6.29 | 6.50 | 7.93 |
| GNN (parameter $\times 1.66$) | 37.58 | 30.61 | 30.05 | 6.34 | 6.49 | 7.90 |
| **MCG (parameter $\times 1$)** | **71.56** | **59.17** | **49.58** | **14.65** | **14.06** | **13.28** |

# C    Extended Related Work

## C.1    World Models

Contemporary research on world models can be delineated along two principal trajectories, each characterized by fundamentally divergent objectives. The first research trajectory conceptualizes world models as instrumental components, primarily serving either as predictive mechanisms to facilitate planning processes or as training apparatuses for policy optimization. Conversely, the second research trajectory approaches world models as generative frameworks, with the explicit objective of predicting future environmental states with high fidelity.

Within the first trajectory, numerous approaches leverage world models as predictive mechanisms to facilitate planning processes. Notable exemplars include methodologies employing Monte Carlo Tree Search (MCTS) to identify optimal action sequences [65, 12, 17] and techniques utilizing cross-entropy methods to efficiently sample continuous actions [25]. Complementary to these, a substantial corpus of research utilizes world models to augment policy learning, wherein the learned dynamics models either provide supplementary supervision signals or function as synthetic environments for policy optimization. This subcategory encompasses seminal works such as Dreamer [22, 23, 24], SimPLe [38], IRIS [53], $\Delta$-IRIS [54], and DART [1]. These methodologies predominantly employ model-based reinforcement learning frameworks, wherein learned dynamics models generate synthetic data to facilitate policy model training.

The second trajectory is predominantly focused on the development of sophisticated generative models designed to predict future environmental states with high verisimilitude. Representative examples include DIAMOND [3], Navigation world models [4], Oasis [11], and The Matrix [16]. These models are engineered to capture the underlying stochastic dynamics of complex environments and generate realistic future states conditioned on current states and selected actions.

Notwithstanding these advancements, it is imperative to acknowledge that methodologies relying predominantly on statistical correlations frequently exhibit performance degradation when confronted with distributional shifts in environmental conditions, thereby compromising their capacity for robust generalization across diverse scenarios.

## C.2    Causal Discovery for World Models

Causal discovery offers a rigorous analytical framework for addressing the inherent challenges associated with distributional shifts in world models. By elucidating and exploiting causal relationships rather than mere statistical correlations, these methodologies significantly enhance both the interpretability and generalization capabilities of learned models, thereby facilitating more robust decision-making processes in complex, non-stationary environments.

The methodological landscape of causal discovery can be taxonomized into two principal categories: constraint-based approaches and score-based approaches. Constraint-based methodologies, exemplified by the PC algorithm [68], employ conditional independence tests to systematically infer causal relationships among variables. Conversely, score-based approaches utilize statistical evaluation metrics to assess the plausibility of alternative causal structures. Prominent instantiations include the Greedy Equivalence Search (GES) [51], which implements a greedy search algorithm to optimize a predefined scoring function, and methods leveraging the Bayesian Information Criterion (BIC). Both methodological paradigms endeavor to recover causal graphs by identifying the Markov equivalence class of the underlying causal structure. Nevertheless, it is imperative to acknowledge that the causal graph learned through these approaches is not uniquely identifiable, as multiple distinct causal architectures can manifest identical conditional independence relationships.

To mitigate this fundamental identifiability challenge, contemporary advancements in causal discovery have increasingly focused on incorporating interventional data to enhance the discriminability of causal structures [69, 52, 33, 72, 6, 66, 37]. Despite these methodological innovations, a significant limitation persists: these approaches frequently operate under the restrictive assumption that the underlying causal graph remains temporally invariant throughout the learning process. This stationarity assumption may be violated in dynamic environments, wherein causal structures can evolve temporally due to myriad factors.

## C.3  Comparison with FCDL

**Motivation and Comparison to Context-Dependent Causality.**    The initial motivation for our work came from a fundamental question: Why do causal discovery methods often fail in open-ended environments?

We observed that causal models aim to describe the rules of the world. However, in open-ended settings (e.g., open-ended games, multi-agent systems, or LLM-based dialogue), no single observation window can capture all possible events. This leads to instability: a causal relation valid in one context may fail in another. For example, Newtonian physics holds at the macroscopic level but breaks down at the quantum scale. FCDL also emphasizes this issue (but it's not the first one), arguing that a complete causal law must account for variation across different contexts. Therefore, our high-level motivation aligns with them. many previous works have explored this motivation context-dependent causal structures.

However, FCDL is not the first one discusses about context-dependent causality. The idea of context-dependent causality was formally introduced by earlier Huang et al. [29], and also discussed earliest in Chapter 10 of Pearl's Causality [56].

**Exploration and the Limitation of Passive Learning.**    A key challenge in scaling up to open-ended worlds is how to explore effectively. Relying solely on random exploration or limited observed data is insufficient to discover hidden causal mechanisms. For instance, in scientific discovery, new causal knowledge often emerges only after targeted investigation of paradoxes. Similarly, in dialogue agents, causal relationships may even reverse depending on the context. Unlike prior works including FCDL and Huang et al. [29], which rely solely on observation passive data, our method explicitly considers the unobserved causal space, and how newly explored data may lead to shifts in both the causal graph and the context boundary.

**Causality-Seeking Agent.**    We aims to go beyond passive modeling to active exploration for causal discovery. Our curiosity and intervention-based strategy address a core limitation of data-driven methods: the inability to discover unknown causal structures. Our primary theoretical contribution is the construction of a formal intervention-based meta-causal graph, with associated theorems and learning framework.

## C.4  Connections to Model-Based RL and Probabilistic Graphical Models

Our framework can be viewed as a special case of model-based RL (MBRL): we learn a world model and use it to plan. The difference lies in the causal structure we learn and use. Classical factored or relational MBRL methods learn structured predictive models [20, 42, 60], which improve sample-efficiency but typically remain correlational. By contrast, we explicitly model a Meta-causal graph whose edges carry interventional semantics (actions are treated as interventions), and we pair structure learning with intervention design and identifiability analysis for context-dependent mechanisms (meta states).

From the viewpoint of probabilistic graphical models (PGMs), our method instantiates a structural causal model (SCM) with a discrete context variable $u$ (the meta state): each context selects a graph $\mathcal{G}_u$ and its causal skeleton $M_u$, while the decoder $D$ outputs a probabilistic mask $\hat{M}_u$ that we discretize via Gumbel reparameterization. This preserves the graphical factorization of a PGM but endows it with *interventional* semantics in the sense of Pearl's do-calculus [56]. Practically, this differs from standard PGM-based world models by (i) targeting *causal* edges rather than solely predictive factors, (ii) using curiosity-driven interventions to break equivalence classes, and (iii) handling changing mechanisms via meta states with an adaptive, minimal codebook.

Finally, our curiosity module is compatible with latent-variable exploration used in POMDP-style MBRL: intrinsic objectives (entropy, feature discrepancy, predictive-NLL, accuracy) can be plugged in as information-seeking criteria, but here they are *aimed at* reducing uncertainty over *causal structure* (meta states and edges), not only over hidden state trajectories.

# D  Theoretical Connections between Meta-Causal Graphs, Open-Endness, and Gödel Machines

Open-endedness is defined in terms of continuous novelty and learnability, highlighting its significance for creating artificial superhuman intelligence (ASI) [30]. Additionally, the Gödel Machine concept proposed encapsulates self-referential improvement mechanisms through formal proof-driven code rewriting. Here, we analyze the theoretical relationships between the MCG framework, open-endedness, and the Gödel Machine [63].

## D.1  Open-Endness and Meta-Causal Graphs

The Meta-Causal Graph framework establishes connections with open-endedness and self-improving systems. Open-endedness is characterized by continuous generation of novelty within comprehensible boundaries, a foundational requirement for advanced artificial intelligence systems. Our MCG framework demonstrates key open-ended properties in several fundamental ways:

First, the curious causality-seeking agent operationalizes open-ended exploration through a curiosity-driven intervention strategy. By maximizing entropy-based reward signals focused on regions of causal uncertainty, the agent persistently generates novel interventions and discoveries. This mechanism directly addresses the novelty requirement of open-endedness, as the agent autonomously uncovers and probes previously uncharted causal relationships and latent meta states.

The MCG framework maintains interpretability by organizing new causal knowledge within an explicit meta-causal graph. As new mechanisms are discovered, they are systematically integrated, ensuring ongoing comprehensibility despite growing complexity. Vector quantization further enables efficient and semantically clear assignment of novel observations to distinct meta states.

The MCG framework thus marks a step toward genuinely open-ended learning systems capable of autonomously exploring, understanding, and adapting to complex, dynamic environments with evolving causal structures.

## D.2  Gödel Machines and Meta-Causal Graphs

It is important to note that while the MCG framework does not implement Gödel Machine-style code-level self-rewriting, its core mechanism nevertheless embodies a form of self-improvement focused on its own knowledge structure (the Meta-Causal Graph). The curious causality-seeking agent, driven by curiosity-based rewards, actively collects new data and, upon encountering prediction failures or high uncertainty, dynamically refines and expands its set of causal subgraphs and meta state mappings. This process ensures the agent can continually update and improve its world model in open environments, achieving self-correction and knowledge-level self-evolution.

# E  Implementation Details

## E.1  Meta-Causal Graph Learning from Experience

We learn causal subgraphs under latent meta states from agent experience. Following FCDL's pioneering use of the VQ-VAE architecture for causal-graph discovery [31], we adopt the original VQ-VAE framework of van den Oord et al. [74] to learn the causal subgraph for each meta state. Specifically, the process contains two steps: (1) identifying the latent meta state, which activates a specific causal subgraph, and (2) learning the subgraph from the agent's explored experience data.

**Identifying Latent Meta State:** We embed the observed state and assign it to a corresponding meta state $u \in U$ by vector quantization as follows. We define the meta state assignment as $C(x) = \arg\min_{u \in U} \left\| E(x) - z_u \right\|_2^2$, where $E \colon \mathcal{X} \to \mathbb{R}^d$ is an encoder mapping the observed state $x$ to a $d$-dimensional embedding, and $\{z_u\}_{u \in U} \subset \mathbb{R}^d$ is a learnable codebook of prototype embeddings, each representing a distinct meta state.

**Subgraph Learning:** After obtaining the embedding $z_u$ for a given state, we employ a decoder network $D \colon \mathbb{R}^d \to [0,1]^{p \times p}$ to predict a probability matrix $\hat{M}_u$, which estimates the underlying causal skeleton matrix $M_u$ of $\mathcal{G}_u$. Each entry $\hat{M}_u[i,j]$ denotes the probability that the directed edge

$i \to j$ is present in causal subgraph associated with meta state $u$. We then sample each $M_u[i, j]$ from Bernoulli($\hat{M}_u[i, j]$) using the Gumbel-Softmax reparameterization [35, 50], enabling end-to-end gradient-based learning of the discrete structure. To update the codebook, we use the following update rule:

$$\mathcal{L}_{\text{quantization}} = \|\text{sg}(E_\phi(X)) - z_u\|_2^2 + \beta \|\text{sg}(z_u) - E_\phi(X)\|_2^2,$$

where $\text{sg}(\cdot)$ is the stop-gradient operator. The first term is the reconstruction loss and the second term is the commitment loss to avoid the output of the encoder growing arbitrarily [74].

