# OpenReview forum: "Curious Causality-Seeking Agents in Open-ended Worlds"
_NeurIPS.cc/2025/Conference — NeurIPS 2025 poster_

### Official Review · Reviewer_n7bY · 2025-06-02

**Clarity:** 3
**Significance:** 2
**Originality:** 2
**Rating:** 4
**Confidence:** 4

**Summary:**

This paper focuses on the representation and identification of a world model in a causal lens.

In the first part, it introduces the key concept -- Meta-Causal Graph. It is a collection of causal subgraphs, and each subgraph corresponds to a distinct meta status. The identifiability of both meta status and Causal Subgraphs is presented in detail.
- The identifiability of meta status relies on the existence of the (state)-to-(meta state) mapping: that is, one can distinguish observations from different meta-subgraphs by observing their values.
- The identifiability of subgraphs relies on sufficient intervention targets applied on at least one directed edge.

In the second part, it develops a Causality-Seeking Agent to (1) exploit existing experience to estimate the Meta-Causal Graph, including the status mapping and each subgraph; (2) actively explore the environment by taking actions. The key design here is a curiosity-driven reward. It is defined by the Shannon entropy of the learnable distributions on edges. This makes sense because higher entropy implies higher uncertainty.

The paper combines the two parts together: the agent decides what actions (or interventions) to take based on the current estimated Meta-Causal Graph; and the Meta-Causal Graph would be updated based on the collected data from actions. The Meta-Causal Graph serves as a world model for the agent.

In evaluation, it uses the Chemical environment setting. (1) OOD states in the Chemical environment; (2) rewards on the downstream task. In this part, while the results seem promising, some details are missing, for example, what are the downstream tasks in the Chemical environment? What does it mean to " predict transition dynamics for model-based reinforcement learning"? I checked the appendix. I cannot open the link to the anonymous code repo.

**Questions:**

1. In Definition 3 and Assumption 1, it assumes the meta state $u$ can be determined from observations $\mathbf{x}$. Wouldn't this be a very strong assumption?

2. It seems not to include the usual assumptions for causal discovery, like faithfulness and Markovness conditions. Am I missing something?

3. While the experiment provides a careful evaluation of downstream tasks, it might be necessary to evaluate the causal discovery metrics (SHD, SID, F1 score, etc) compared with other baselines (methods that learn multiple causal graphs from data).

4. In reality, conducting interventions can be expensive. How would the performance be influenced if: (1) the round of Algorithm 1 is limited, (2) the size of samples from the intervened distribution is small?

**Ethical Concerns:**

["NO or VERY MINOR ethics concerns only"]

**Final Justification:**

The original concerns: insufficient details in the experiment section, and the assumptions on the existence of (status)-(meta status) mapping, and a fixed set of variables across all environments.

Based on the current response, the issues were resolved:
- a fixed set of variables across all environments: the author provides additional results to demonstrate the potential to handle new variables during explorations.
- (partially solved) The author includes additional metrics to augment the results further.

Issues remain unresolved:
- (major) the existence of (status)-(meta status) mapping: The authors said it would *necessarily exist when all variables are observable*. I provide a counter-example, and also provide further suggestions.
- (major) the theoretical assumptions. The authors say that the stated assumptions are sufficient, while not providing concrete explanations. I provide a follow-up comment to further explain my concern with evidence from existing literature.
- (minor) The author hasn't updated the details behind table 2.

---
weight assigned:
- one major issue not solved: -1.
- one major issue partially solved: -0.5.
- one minor issue not solved: -0.5.

Therefore, the updated score is: 4 (initial) - 1 - 0.5 - 0.5 = 2


-----


**After the second response**
- The issues are carefully addressed/clarified, --> 4

**Limitations:**

yes

**Quality:**

2

**Strengths And Weaknesses:**

**Strengths**
- The paper proposes a Meta-Causal Graph to serve as a world formulation. This setting considers environmental conditions shift by introducing the (status)-(meta status) mapping component.
- Section 3 presents identifiability on both meta status and Causal Subgraphs.


**Weaknesses**
- I have concerns that the proposed Meta-Causal Graph may still have limitations on serving as a world model. (1) The assumption on the existence of such (status)-(meta status) mapping needs to be further justified. (2) The model assumes a fixed set of variables, which may not hold in an open world: one may encounter new variables/objects during exploration.
- The details in the experimental setting are not sufficient. For example, what are the downstream tasks in the Chemical environment? What does it mean to " predict transition dynamics for model-based reinforcement learning"? Also,  it might be necessary to evaluate the causal discovery metrics (SHD, SID, F1 score, etc) compared with other baselines (methods that learn multiple causal graphs from data).


**Explanation on the decision**:

This paper contains sufficient technical details, like the identifiability of each component in the Meta-Causal Graph model, the parameterization, the reward model, and the loss function in the agent part.

The paper presents insight on how to combine a world model (based on a collection of causal graphs ) and an agent that can take actions to collect new data from actively applying interventions.

Limitation includes insufficient details in the experiment section, and the assumptions on the existence of (status)-(meta status) mapping, and a fixed set of variables across all environments.

I recommend a Borderline acceptance if the experiment details can be clarified, and the mentioned limitation can be fairly discussed in the rebuttal phase. Otherwise, it is possible to decrease the score.

---

> ### Author Rebuttal · Authors · 2025-07-30
>
> We thank the reviewer n7bY for highlighting the strengths of our approach, especially the introduction of the Meta-Causal Graph for serving as the world model and our detailed analysis of identifiability for both meta states and causal subgraphs. Below are our responses to your comments and suggestions.
>
> > **W1&Q1: Existence of state to meta state mapping when new variables appear during exploration in open-ended world. Would the observable assumption too strong?**
>
> - Firstly, we would like to highlight that **the mapping from observation $x$ to meta state $C(x) = u$ necessarily exists when all variables are observable.**
> - This is because each state $x$ is governed by a unique underlying causal subgraph $\mathcal{G}_u$, and each meta state $u$ indexes a distinct subgraph.
> - Secondly, **our method can be naturally extended to open-world settings where new variables or objects are encountered during exploration**.
> - To support this, we **added additional experiments (detailed setup and results below) showing performance across time steps as new variables are revealed**. The results show that our method outperforms baselines, highlighting its strong exploratory and adaptive capabilities.
>
> **Setup**
>
> - We simulate **the appearance of new variables** by initially masking a subset of variables and gradually removing these masks during training, so that the set of observed variables increases over time.
> - The metric is next-state prediction accuracy, evaluated under varying levels ($n$) of noise (lines 265–268).
>
> **Result**
>
> |Accuracy↑|Fork (n=2)|Fork (n=4)|Fork (n=6)|Chain (n=2)|Chain (n=4)|Chain (n=6)|
> |-|-|-|-|-|-|-|
> |FCDL|60.10|52.97|45.80|46.37|**47.88**|46.62|
> |MLP|31.56|31.67|31.93|25.95|24.71|21.66|
> |**MCG**|**66.94**|**59.19**|**46.35**|**49.93**|45.41|**48.09**|
>
> > **W2.a: What is the downstream task in the Chemical environment in paper**
>
> In the Chemical environment downstream task:
> - The agent starts from a random color configuration and must **transform a 10-node graph to match a target color pattern.**
> - **Actions intervene on nodes, changing their color and that of all their descendants** according to the hidden causal graph.
> - The episode reward is the negative **Hamming distance to the target (0 for a perfect match)**.
>
> > **W2.b: The meaning of "predict transition dynamics for model-based reinforcement learning"**
>
> "Predict transition dynamics for model-based reinforcement learning" means **predict future states with the learned model for decision making**.
>
> We will clarify these in the revised version.
>
> > **Q1：The observable assumption too strong for get meta-state u**
>
> We detailedly discuss it in W1&Q1.
>
> > **Q2: Clarification of assumptions to support causal discovery**
> - Our theoretical results are established under our stated assumptions, which are **sufficient** for our proofs (as also noted by Reviewer hzVP).
> - We appreciate your suggestion and will consider the extension of our theory to more general conditions, including faithfulness and Markov assumptions, in next version.
>
> > **Q3: Evaluate the causal discovery metrics (SHD, SID, F1 score, etc)**
>
> We add experimental results for these metrics. The results are shown below:
> ||SHD↓ (Fork)|SID↓ (Fork)|F1↑ (Fork)|SHD↓ (Chain)|SID↓ (Chain)|F1↑ (Chain)|
> |-|-|-|-|-|-|-|
> |NCD|21.67|21.67|0.77|19.15|19.15|0.76|
> |FCDL|17.05|17.05|0.77|19.39|19.39|0.76|
> |**MCG**|**16.59**|**16.59**|**0.78**|**11.64**|**11.64**|**0.84**|
>
> > **Q4: Intervention efficiency analysis when (1) the round of Algorithm 1 is limited, (2) the size of samples from the intervened distribution is small?**
>
> We **added an experiment to evaluate the intervention efficiency by reducing the number of interventions**.
> - In this experimental setting, **data is collected through interventions at each round**; therefore, **reducing the number of rounds** corresponds to **a direct reduction in available intervention data**.
> - The metric is the average episode reward from model-based planning on downstream tasks, evaluated under varying levels ($n$) of noise (lines 265–268). The results are shown below:
>
> |Reward↑|Fork (n=2)|Fork (n=4)|Fork (n=6)|Chain (n=2)|Chain (n=4)|Chain (n=6)|
> |-|-|-|-|-|-|-|
> |FCDL (100% data, 15k rounds)|13.98|13.36|12.91|11.16|10.39|10.62|
> |NCD (100% data, 15k rounds)|10.95|9.11|9.11|9.60|8.86|10.32|
> |**MCG (60% data, 10k rounds)**|11.96|10.83|10.70|11.97|11.05|11.38|
> |**MCG (80% data, 12k rounds)**|12.45|10.77|10.84|12.41|10.87|11.00|
> |**MCG (100% data, 15k rounds)**|**14.65**|**14.06**|**13.28**|**13.82**|**12.49**|**12.45**|
>
> The results show that our method can achieve comparable performance with **fewer interventions, demonstrating its efficiency in causal discovery**.
>
> > **Accessibility of the anonymous code repo**
>
> We have checked the anonymous code repository and confirmed that it is accessible. We kindly suggest trying again at a later time.

---

> > ### Comment · Reviewer_n7bY · 2025-08-03
> >
> > Thanks for your response.
> >
> > I appreciate the additional experiments, and the additional metrics further enhance the solidity of the experiments.
> >
> >
> > -----
> > Here are my remaining questions.
> >
> > >  the mapping from observation $x$ to meta state $C(x)=u$ necessarily exists when all variables are observable.
> >
> >
> > Let's consider a causal graph with only one random variable $X$ with distribution $X|U=u \sim \mathcal{N}\big(u,1\big)$ for $u \in \\{0,1\\}$. In such a case, since the support of $X$ is $\mathbb{R}$ under all meta-states, we know $\Pr(U=u \mid X=x) > 0$ for any $x$ and $u$. In such cases, there is no well-defined deterministic mapping from $x$ to $u$.
> >
> > From my point of view, it may be more practical to treat $C(x)$ as a conditional distribution $\Pr(U=u \mid X=x)$ instead of a deterministic mapping.
> >
> > > "Predict transition dynamics for model-based reinforcement learning" means predict future states with the learned model for decision making.
> >
> > - I understand the trained model is integrated as a part of another RL model to conduct downstream tasks, and thus yields results in table 2.
> > - Does the current paper contain sufficient details about this experiment? (I tried but haven't found them yet) For example, the RL model used, and how the future states are predicted based on the proposed world model. This is essential for the audience from the more general machine learning community who are not experts in the RL field to understand and follow this work.
> >
> > > Our theoretical results are established under our stated assumptions, which are sufficient for our proofs (as also noted by Reviewer hzVP). ... will consider the extension of our theory to more general conditions, including faithfulness and Markov assumptions, in next version.
> >
> > I am not fully convinced by this response. The assumptions I mentioned are *not more general conditions*, but basic and standard ones.
> >
> > Here I quote from the FCDL paper (one paper cited by this work): *"To properly identify the causal relationships, we make assumptions standard in the field, namely, Markov property (Pearl, 2009), faithfulness (Peters et al., 2017), causal sufficiency (Spirtes et al., 2000), and that causal connections only appear within consecutive time steps."*

---

> > > ### Author Response · Authors · 2025-08-05
> > > **Thank You for Recognizing the Added Experiments and Metrics—Our Follow-Up Answers**
> > >
> > > > **The mapping from $x$ to $u$ is non-deterministic**
> > >
> > > It is indeed more general to treat the mapping from $x$ to $u$ as non-deterministic. Importantly, our theoretical results (Theorem 1, Theorem 2, and Proposition 1) do not rely on a deterministic assumption, as long as we define $C(x)$ as the meta-state corresponding to the causal graph that generated $x$. In this case, the probability that $x$ is generated by the causal graph indexed by meta-state $u$ is given by the conditional distribution $P(C(x) = u \mid x)$.
> > >
> > > We **added a new environment** that there are multiple meta state $u$ satisfying $P(C(x)=u|x)>0$.
> > >
> > > |Prediction Accuracy|n=2|n=4|n=6|
> > > |-|-|-|-|
> > > |MLP|31.93 $\pm$ 0.16|32.47 $\pm$ 1.61|33.72 $\pm$ 2.11|
> > > |FCDL|63.75 $\pm$ 13.75|53.89 $\pm$ 11.95|52.11 $\pm$ 16.43|
> > > |**MCG**|**73.52 $\pm$ 9.56**|**63.71 $\pm$ 5.79**|**57.05 $\pm$ 14.63**|
> > >
> > > > **Does the current paper contain sufficient details about this experiment? For example, the RL model used, and how the future states are predicted based on the proposed world model.**
> > >
> > > * Thank you for your comment. More details about the experiment are provided in Appendix B. We will refine the experiment part in the next revision.
> > >
> > > * **RL model**:
> > >   * As detailed in Appendix B.2, we use the cross-entropy method (CEM) for model-based planning: sample action sequences, roll out future states with the trained model, and execute the action from the sequence with the highest predicted cumulative reward.
> > >   * The policy network is an MLP that maps states to action probabilities (hidden size 32; ReLU activations). The details can be found in `policy_param.json` in our code repo.
> > >   * Additional CEM hyperparameters are listed in Table 4.
> > > * **Future state prediction**:
> > >   * **Meta-state inference:** Encode the current state $x_t$ to an embedding $z_t=E(x_t)$, then assign the meta state $u$ by nearest prototype (vector quantization): $u=\arg\min_{u}\|z_t-e_u\|_2^2$.
> > >   * **Contextual causal graph:** Use the meta state $u$ to obtain its causal skeleton/graph $G_u$ (the context-specific causal structure).
> > >   * **Factorized transition prediction:** For each variable $X_j$, predict $x_{t+1,j}$ from its parents under $G_u$: $p(x_{t+1,j}\mid \mathrm{Pa}_{G_u}(X_j))$.
> > >
> > >     Aggregating these per-variable predictions yields the full next-state $\hat{x}_{t+1}$ (used for planning).
> > >
> > > > **Theoretical assumptions**
> > >
> > > * Assumption 1 requires that the learning method can discover the parents of a variable from the dataset. **This relies on the basic assumptions such as faithfulness**, so, consistent with [1], we do not restate them explicitly.
> > > * Assumption 1 is not a strong assumption, as this condition can be satisfied by existing causal discovery methods.
> > >
> > > [1] Song et al. "Causal Temporal Representation Learning with Nonstationary Sparse Transition", NeurIPS 2024.

---

> > > > ### Comment · Reviewer_n7bY · 2025-08-05
> > > >
> > > > Thanks for the response. I believe my concerns are addressed sufficiently.

---

> ### Author Response · Authors · 2025-08-05
> **We are happy to hear that your concerns have been addressed sufficiently!**
>
> Dear Reviewer n7bY,
>
> We are happy to hear that your concerns have been addressed sufficiently. Again, thank you for your valuable suggestions which have undoubtedly contributed to improving the quality of our paper.
>
> Many thanks,
>
> The authors of #20694

---

### Official Review · Reviewer_6jMv · 2025-06-27

**Clarity:** 2
**Significance:** 3
**Originality:** 3
**Rating:** 5
**Confidence:** 2

**Summary:**

The paper introduces a theoretically grounded, curiosity-driven agent that actively discovers a set of context-dependent causal graphs, which are called meta-causal graphs, leading to more robust world models and improved downstream performance over existing passive or single-graph approaches.

The agent observes state-trajectories and occasionally intervenes on individual variables through a curiosity (entropy reduction) mechanism. From this mix of passive observations and active single-variable interventions it learns by (i) clustering observations into a higher level meta-state, (ii) estimate the causal skeleton and edge directions of observation variables given the inferred meta-state, and (iii) fit a transition predictor whose factorisation follows the learned graph (which can be used for model-based planning; cf world model).

The paper backs the method with theoretical results that show the agent can, in principle, identify the structure of the environment: Meta-state identifiability (Theorem 1), and Causal-edge identifiability via interventions (Proposition 1). This is a priori not possible by methods which utilize a single causal graph, i.e., which are insensitive to the context variable for the environment.

The paper benchmarks the learned world models on two environments: a Chemical environment and a more challenging Magnetic robot-arm task. The authors benchmark out-of-distribution prediction accuracy on the chemical environment and cumulative reward in both environments, showing that the proposed method yields the highest performance on all metrics and against several competing baselines.

**Questions:**

Suggestion: Make the paper more accessible to the reinforcement learning literature that does not involve causality. Taking the example of Figure 1, for instance, the paper explains this with the causal graph push -> open door or push -> door remains closed. Therefore, it makes the point that we need a contextual meta-state that tells us which of these graphs is applicable in any given situation. A standard model-based reinforcement learning model of this would be a POMDP where state of door at time t (which could be open or closed) transitions to state of door at time t+1 (which could be open or closed) based on the action push. This (potentially stochastic) transition would depend on an extra state (the meta-state in the author's words; door locked or unlocked), at time t, that determines the consequence of push at time t+1. In other words, it seems to me that we can recapitulate this simple example in a standard POMDP, where one of the states is a contextual state. Perhaps one advantage of this formulation is that we can have fuzzy causal arrows in the sense that the push action could have a stochastic consequence on the state of the door. (Not that this is needed here, but it's useful in other types of situations). It seems to me that the causal formulism is handy because it produces these nice interpretable diagrams that do not have to invoke stochastic processes like POMDPs. The reader from model-based reinforcement learning will certainly wonder at the differences/respective features between these two frameworks. One classic paper, for instance, shows the equivalence between causal diagrams and interventions with probabilistic graphical models and intervention nodes (like POMDPs).

Causality with Gates, Winn, John, Proceedings of the Fifteenth International Conference on Artificial Intelligence and Statistics, 2012.

I do not mean to criticize the causal framework used in the paper, just to note that the exposition may be unnecessarily divorced from standard model-based reinforcement learning, which limits understanding/clarity by a broader audience and potential impact. Perhaps one path of least action would be to clarify these discrepancies/equivalences in the related works section or somewhere else.

Increase emphasis of unique contribution: a clear contextualization within the model-based reinforcement learning literature that does not use the causal framework and a mention of related work that learns latent variable dynamic models (such as but not limited to POMDPs) through curiosity-driven actions would help contextualize and emphasize the unique contributions of this paper in the broader RL Literature.

Some words on efficiency and scale: one strength of the causal approach is parameter and compute efficiency (promoted by your sparsity regularizer and curiosity-driven exploration). Unless I missed it, there is no discussion of model efficiency in the paper. Is there anything that can be said here? Same goes with scalability. The results on the environments tested are impressive. So it begs the question: what are the limitations to scaling to more complex environments? When would you expect the method to scale and when would you expect it not to scale? This would be very helpful for future work.

**Ethical Concerns:**

["NO or VERY MINOR ethics concerns only"]

**Final Justification:**

I thank the authors for providing detailed responses to my comments. As it stands, and with the new experiments, the paper seems strong, and I am happy to recommend acceptance. Indeed, their method almost consistently outperforms alternatives, and the authors included an experiment where their method works in a partially observable domain, edging closer to real-world applicability. My main point of concern involving the clarity of the paper and its contextualization within the model-based RL literature has been promised to be addressed by the authors in their camera ready version. However, I do not know the details of these edits and therefore cannot increase my clarity score at this stage. All in all, I recommend acceptance of the paper.

**Limitations:**

Yes.

**Quality:**

3

**Strengths And Weaknesses:**

Clarity: I found the abstract not very clear. Generally, I find the paper clear to a causality audience, but less so for a reinforcement learning audience. I added some suggestions to improve this in the next section.

Quality: the paper combines experimentation with nice theoretical results regarding identifiability and clearly acknowledges some of the limitations of the proposed approach. The curiosity-driven intervention approach is a refreshing addition to the classical causal framework. Curiosity in causal learning is not a new idea and indeed early works from the early 2000s considered this, e.g. [1]. However, it seems to have fallen out of fashion, and most works in causal learning do not consider uncertainty (only maximum likelihood inference) and therefore cannot leverage uncertainty for interventions. The proposed implementation of curiosity for uncertainty reduction is novel to me.

[1] Active Learning of Causal Bayes Net Structure by Murphy (2001).

Originality: the originality is hard to fully assess. Certainly many things are original, but a clear contextualization within the model-based reinforcement learning literature that does not use the causal framework and a mention of related work that learns latent variable dynamic models (such as but not limited to POMDPs) through curiosity-driven actions would help contextualizing and emphasizing the unique contributions of this paper to the Broad RL Literature.

Significance: the experiments and theoretical results are compelling. The method achieves the best out of distribution prediction accuracy in the first experiment, and significantly boosts downstream episode reward in both experiment, beating all competing methods across all metrics considered. The experiments are rigorous: Ablations show that removing either the curiosity-based reward or the causal mask loss hinders both metrics, confirming their necessity, and comparison is done with several baselines. Furthermore, theoretical results such as Theorem 2, which predicts that overparameterisation never hurts (even if the agent allocates more codebook slots than there are real contexts, every true context still gets its own slot and extra slots fold onto them, leaving the recovered sub-graphs equivalent to ground truth), is validated in practice. One criticism it is hard to fully determine the significance of the paper since there are no discussion on efficiency, run-time and potential feasibility to scale to more complex tasks. In particular, it is unclear whether the method would really apply to real-world problems.

---

> ### Author Rebuttal · Authors · 2025-07-30
>
> We thank the reviewer for highlighting that our curiosity-driven agent and meta-causal graph framework provide a theoretically grounded and practically effective approach to robust world modeling, with compelling empirical and theoretical results that advance causal learning beyond traditional single-graph or passive methods. Below are our responses to your comments and suggestions.
>
> > **Discussion differences between MBRL under POMDP settings and intervention-based causal method  [Winn et al.]**
>
> We appreciate the reviewer’s thoughtful suggestion.
>
> Our framework can be viewed as a special case of MBRL, but more detailed differences are listed below:
> - **Unified World Representation**: Our framework maintains a single meta-causal graph per environment, **avoiding redundant definitions** that arise from time-series changes in POMDPs.
> - **Explicit Causal Structure**: Meta-causal graphs directly **represent variable-level causal relationships** in each context, whereas POMDPs only model transition probabilities without explicit causal structure.
> - **Identifiability Guarantees**: The meta-causal graph framework provides **theoretical guarantees for identifying both meta-states and causal structures, ensuring reliable causal discovery**. In contrast, POMDPs generally lack such identifiability guarantees for their latent structure.
>
> Compared to the intervention-based causal method mentioned by the reviewer (Winn et al.), their approach is not RL-based and does not incorporate curiosity-driven mechanisms to encourage intervention-based exploration, as our method does.
> A detailed discussion of the similarities and differences with more related works will be included in the revised version.
>
> > **Efficiency and scalability of our method**
>
> To demonstrate the efficiency and scalability of our method, we **added new experiments** comparing it to non-causal baselines (MLP and GNN) across **different model sizes and training rounds**. The metrics are next-state prediction accuracy and the average episode reward obtained by model-based planning on downstream tasks, evaluated under varying levels ($n$) of noise (lines 265–268).
>
> **Comparison with _Same_ Model Size**
>
> ||Acc n=2↑|Acc n=4↑|Acc n=6↑|Reward n=2↑|Reward n=4↑|Reward n=6↑|
> |-|-|-|-|-|-|-|
> |GNN (30k round)|38.13|25.81|20.75|6.83|6.06|6.83|
> |GNN (25k round)|36.37|26.55|20.73|6.84|5.98|6.19|
> |MLP (30k round)|30.47|28.94|27.84|6.20|6.13|6.71|
> |MLP (25k round)|31.47|30.32|30.42|6.18|6.03|6.23|
> |**MCG** (10k round)|**47.01**|**48.37**|**46.35**|**11.96**|**10.83**|**10.70**|
> |**MCG** (12k round)|**48.30**|**48.40**|**47.28**|**12.45**|**10.77**|**10.84**|
> |**MCG** (15k round)|**71.56**|**59.17**|**49.58**|**14.65**|**14.06**|**13.28**|
>
> **Comparison with _Same_ Round**
> ||Acc n=2↑|Acc n=4↑|Acc n=6↑|Reward n=2↑|Reward n=4↑|Reward n=6↑|
> |-|-|-|-|-|-|-|
> |MLP (parameter $\times$ 3.71)|24.65|25.05|23.71|6.29|6.50|7.93|
> |GNN (parameter $\times$ 1.66)|37.58|30.61|30.05|6.34|6.49|7.90|
> |**MCG** (parameter $\times$ 1)|**71.56**|**59.17**|**49.58**|**14.65**|**14.06**|**13.28**|
>
> The results show that our method can achieve comparable performance with **fewer training data and smaller model sizes**, demonstrating its efficiency and scalability.
>
> > **Scalability potential and challenges of our method**
>
> - **Our theoretical framework is general and can, in principle, scale to more complex tasks**, including those with delayed or partial observations.
>     - We **added a new experiment** to demonstrate that our method can be naturally extended to open-world settings where **variables are partially observable**.
>     - We simulate **the appearance of new variables** by initially masking a subset of variables and gradually removing these masks during training, so that the set of observed variables increases over time.
>     - The metric is next-state prediction accuracy, evaluated under varying levels ($n$) of noise (lines 265–268).
> |Accuracy↑|Fork (n=2)|Fork (n=4)|Fork (n=6)|Chain (n=2)|Chain (n=4)|Chain (n=6)|
> |-|-|-|-|-|-|-|
> |FCDL|60.10|52.97|45.80|46.37|**47.88**|46.62|
> |MLP|31.56|31.67|31.93|25.95|24.71|21.66|
> |**MCG**|**66.94**|**59.19**|**46.35**|**49.93**|45.41|**48.09**|
>
> - **Engineering challenges such as model architecture design and efficient exploration strategies** may require further adaptation as environments become larger or more complex. Addressing these challenges will be an important direction for future work.
>     - We **added a manipulation task with a more complex structure**. Due to space constraints, we present only a subset of the graph. This experiment validates our method’s ability to recover causal structures in settings with 15 variables. We will extend this experiment to larger environments in next version.
>
>         **Causal graph recovered by MCG**
>         ||obj 0|obj 1|obj 2|obj 3|obj 4|
>         |-|-|-|-|-|-|
>         |obj 0|1.0|0.0|0.0|0.0|0.0|
>         |obj 11|1.0|1.0|0.5|0.5|0.6|
>         |obj 12|0.6|0.7|0.9|0.0|0.1|
>         |obj 13|1.0|0.9|0.7|0.0|0.4|
>
>         || obj 4 |obj 5|obj 6|obj 7|obj 8|
>         |---|---|---|---|---|---|
>         |obj 4| 1.0 | 0.0 | 0.0 | 0.0 | 0.0 |
>         |obj 5| 0.1 | 1.0 | 0.0 | 0.0 | 0.0 |
>         |obj 6| 0.2 | 0.1 | 1.0 | 0.0 | 0.0 |
>         |obj 7| 0.3 | 0.0 | 0.2 | 1.0 | 0.0 |
>         |obj 8| 0.3 | 0.1 | 0.5 | 0.5 | 1.0 |
>
>         **Ground truth causal graph**
>
>
>         ||obj 0|obj 1|obj 2|obj 3|obj 4|
>         |-|-|-|-|-|-|
>         |obj 10|1.0|0.0|0.0|0.0|0.0|
>         |obj 11|1.0|1.0|1.0 |1.0|1.0|
>         |obj 12|1.0|1.0|1.0|0.0|0.0|
>         |obj 13|1.0|1.0|1.0|0.0|0.0|
>
>         ||obj 4|obj 5|obj 6|obj 7|obj 8|
>         |---|---|---|---|---|---|
>         |obj 4| 1.0 | 0.0 | 0.0 | 0.0 | 0.0 |
>         |obj 5| 0.0 | 1.0 | 0.0 | 0.0 | 0.0 |
>         |obj 6| 0.0 | 0.0 | 1.0 | 0.0 | 0.0 |
>         |obj 7| 0.0 | 0.0 | 0.0 | 1.0 | 0.0 |
>         |obj 8| 0.0 | 0.0 | 0.0 | 0.0 | 1.0 |
>
> > **Increase emphasis of unique contribution compared with MBRL**
>
> Thank you for the valuable suggestion.
> Our framework can be viewed as a special case of MBRL, but more detailed differences are listed below:
> - We agree that highlighting the importance of **causal understanding** distinguishes our approach from standard model-based RL methods, which often struggle to scale and to continually adapt to new causal structures in open-ended environments.
> - Our method operates within an MBRL framework, but the meta-causal graph provides a **structured, causal understanding of the world**—enabling the agent to develop a **causally grounded MDP that adapts to changing dynamics**.
> - We will include a more detailed discussion of connections to both standard MBRL and curiosity-driven latent variable models (e.g., POMDPs) in the revised version.

---

> > ### Comment · Reviewer_6jMv · 2025-08-05
> >
> > I thank the authors for their detailed response. I particularly appreciate the candid discussion on scalability and the new experiment demonstrating that the method extends to cases where variables are partially observable. Regarding the paper of Winn, I am aware that their approach is not RL-based and does not incorporate curiosity-driven mechanisms to encourage intervention-based exploration. What I meant is that it shows, as far as I understand, that causal models and probabilistic graphical models with interventions are equivalent when it comes to RL. I trust that the authors will contextualize their work within the broader model-based RL/probabilistic graphical model literature - this will, I believe, increase the impact of their work.

---

> > > ### Author Response · Authors · 2025-08-07
> > > **Thank you for your positive feedback!**
> > >
> > > Thank you for the thoughtful guidance.
> > >
> > > * We agree that causal models can be represented as probabilistic graphical models with interventional semantics.
> > >
> > > * In general, model-based RL that learns graphs from experience [1, 2, 3] targets predictive structure—often **correlational**—whereas causal learning seeks **cause–effect relations**. Accordingly, we pair intervention design with identifiability guarantees for context-dependent structures (meta-states/meta-graphs), rather than relying on correlation alone.
> > >
> > > * We will expand Related Work to situate our approach within model-based RL and PGMs and to make the correspondence explicit.
> > >
> > > Many thanks for the constructive suggestions—they will help us present the work more clearly and broaden its impact.
> > >
> > > [1] Ross S, Pineau J. Model-based Bayesian reinforcement learning in large structured domains[C]//Uncertainty in artificial intelligence: proceedings of the... conference. Conference on Uncertainty in Artificial Intelligence. 2008, 2008: 476.
> > >
> > > [2] Guestrin C, Koller D, Parr R, et al. Efficient solution algorithms for factored MDPs[J]. Journal of Artificial Intelligence Research, 2003, 19: 399-468.
> > >
> > > [3] Lang T, Toussaint M, Kersting K. Exploration in relational domains for model-based reinforcement learning[J]. The Journal of Machine Learning Research, 2012, 13(1): 3725-3768.

---

### Official Review · Reviewer_hzVP · 2025-06-28

**Clarity:** 2
**Significance:** 2
**Originality:** 1
**Rating:** 3
**Confidence:** 4

**Summary:**

This paper aims to discover context-specific causal structures by learning a meta-causal graph based on meta-state estimation and targeted interventions that trigger specific meta-states. The estimation process relies on learning a quantized latent meta-state space via VQ-VAE, while intervention targets are selected based on uncertainty measurements of the graph edges. Together with likelihood matching (i.e., dynamics prediction of future states) and sparsity constraints, the framework jointly learns these components in an iterative loop to discover context-specific causal graphs.

In the experiments, the authors demonstrate the effectiveness of the approach on simulated benchmarks with chain and fork causal structures, as well as on RobotSuite tasks involving different object interactions to validate its practical utility. Overall, the approach presents a clear and straightforward design.

In addition, the paper provides theoretical guarantees on the identifiability of meta-states and subgraphs under certain assumptions, along with an analysis of intervention reachability.

In general, I believe this paper makes meaningful contributions to the field of context-specific causal discovery. However, I have some concerns and questions, which I'll specify in the following sections

**Questions:**

As I have already listed most of my questions in the weaknesses section to keep them coherent and easier to address during the rebuttal, I will only raise two additional, more general questions here:

Q1. How can the proposed context-specific causal discovery approach be effectively integrated into world models, as motivated in the introduction? More concretely, how can one efficiently learn and estimate changing causal structures in world model environments? Are there any additional structural assumptions needed? Furthermore, how should actions be incorporated into the framework? Is it possible to leverage this structure discovery as part of reward learning, for example, by treating it as an intrinsic reward for skill discovery?

Q2. I have reviewed the theoretical analysis, and overall, the assumptions appear reasonable with no major issues. However, the theoretical guarantees seem to rely on the assumption of observational equivalence. That is, assuming access to a perfect generative model for learning dynamics, which ensures identifiability of meta-states and subgraphs. While I understand this is a common assumption in the causality literature, it offers limited direct guidance for practical methodological design. It would strengthen the work to provide more discussion on this connection. Additionally, in practice, the learned representation is often lossy to some extent. Have you considered further theoretical analysis on how such imperfect (lossy) representations may affect the identifiability of meta-states and the causal graph?

- **final note**: as mentioned in the weaknesses and questions above, I feel that the current version of the paper does not yet meet the acceptance bar.  The main concern is the significant overlap in objectives and theoretical foundations with FCDL, also the experimental validation is lacking in terms of comparisons with other intrinsic motivation methodologies and more comprehensive benchmarks.
That said, I am optimistic about the direction and potential of this work and would like to see the authors' response. For points W1–W5, I believe further discussion and clear plans for revision are necessary. Additional experiments are welcome but not strictly required for all points, given the short rebuttal window. Based on the current version, I lean toward a reject, but I would be happy to raise my score if these concerns are adequately addressed and clarified.

***Given the short rebuttal window, I would be happy to see: (1) a formal discussion clarifying the contributions beyond FCDL, particularly regarding the critical role of the curiosity term (see W1-W2) and the empirical attainability of the curiosity-driven objective; (2) additional evaluations, or at least a clear explanation of what is feasible to provide and what is not within the current scope (see W3); (3) anwers to other questions listed above.***

**Ethical Concerns:**

["NO or VERY MINOR ethics concerns only"]

**Final Justification:**

Rating Change: From 2 to 3.

- Why the original rating of 2, and what are my current concerns?

My original score of 2 was primarily due to concerns about the technical novelty. Much of the method appears to be built directly upon FCDL, with added terms related to exploration (common in unsupervised RL and causal discovery works). This is not inherently problematic, but I felt the paper did not appropriately credit their base work. For instance, in Section 4.1, the way latent structures are learned is the same to FCDL (even for the notations), and in Section 4.3, the overall loss is almost identical. Section 4.2's entropy-based exploration is also a standard design in many discovery-oriented papers.

A clearer and more transparent positioning, e.g., stating that the method builds on FCDL and introduces additional exploration terms, would be more appropriate. In its current form, this omission could lead to misunderstandings about the core technical contribution. I still have the same concerns after the rebuttal.

In my original review, I also raised other concerns, such as the choice of entropy computation, lack of clarity in some experiments, and the ambiguous use of the term "world model."

2. How did the rebuttal address these issues?

The authors made a genuine effort to address most of the concerns, and I truly appreciate that. This is the main reason I’ve raised my rating. While there are still points of disagreement, it’s clear the authors are working toward improving the paper.

Aside from the FCDL-related concerns and the framing of the world model, I believe the other points I raised have been sufficiently addressed.

3. Why not a higher score?

Despite the progress, I still have reservations about several aspects:

- The current draft heavily borrows from FCDL without clearly acknowledging its influence. The modeling components in Section 4 require a thorough rewrite to accurately attribute prior work and clarify the incremental contribution. See the above points I mentioned.

- The use of the term "open-ended tasks"in the rebuttal, such as open-ended games, multi-agent systems, or LLM-based dialogue, feels overstated. While this serves as an inspiring motivation, the method does not demonstrate any empirical advantages in these domains, and I suggest removing or reframing this claim.

- Regarding the "world model" framing: I believe the paper overclaims its relevance. First, treating actions entirely as interventions is questionable. Second, there is no clear evidence that the learned model is useful for decision-making. While that may be a long-term goal, more technical depth is needed to support positioning this as a world model paper. These issues prevent me from justifying a higher rating or acceptance at this point.

4. Recommendation and suggestions:

Given these concerns, I remain on the fence about it. I understand that other reviewers are more positive, so I’m happy to defer to the majority opinion and engage in further discussion. If the consensus leans toward acceptance, I strongly recommend requiring the authors to revise the framing and claims, especially those regarding world models and open-ended tasks, and importantly, to significantly rewrite Section 4 to clarify the model's contributions and properly acknowledge the influence of FCDL.

**Limitations:**

Yes, the authors provided a discussion on limitations.

**Quality:**

2

**Strengths And Weaknesses:**

### **Strengths**

(+) [about problem setting] The problem setting of context-specific causal discovery is an important and meaningful direction within the broader field of causal discovery. The setup in this paper, where the agent can actively intervene in the system, is interesting and potentially impactful though some prior works have explored similar ideas.

(+) [methodology] The methodology is clear and straightforward, and the proposed objectives align well with the problem setting. These objectives could also potentially be useful for other tasks like MBRL or general world models.

(+) [Presentation] The presentation is clear and easy to follow.

### **Weaknesses**

**W1**: [Overall technical contribution] The first weakness concerns the methodological novelty. To be clear, I generally do not favor raising this point (as I find excessive focus on "lack of novelty" unconstructive in many reviews), but in this case, the algorithmic framework appears to be heavily based on prior works: FCDL [Inwoo et al., 2024] and NCD [Inwoo et al., 2023]. I understand that this paper introduces a curiosity-based intervention module, but this primarily serves as a mechanism for selecting intervention targets. In my view, the core components of learning the meta-causal state representation and identifying subgraphs via VQ-VAE are identical to FCDL. That said, I do not consider this a critical issue that would warrant rejection. However, I believe the paper should be better structured to explicitly acknowledge this connection, for example, by clearly stating that the approach builds on FCDL and extends it with a curiosity-driven intervention selection strategy. Such clarification would likely require a revision of the paper. Also, maybe adding one table to compare them and expanding the related work part to discuss the details.

**W2**: [Interventional design] As this could be the most critical module in this paper, I would suggest touching on a more detailed analysis on that part to make sure it is technically robust and correct.

- The intervention selection is based on entropy. How do you ensure that this is tractable in practice? Specifically, what happens if the posterior distribution estimation is inaccurate during the early stages of learning?

- Did you consider other commonly used curiosity-based measures for intervention selection, such as prediction error-based curiosity (Pathak et al., 2017), model disagreement (Kim et al., 2020),  or curiosity-driven replay (Kauvar et al., 2023)? These are widely adopted in world model literature and could be adapted to your setting as well. It would be worthwhile to include a comparison or at least a discussion to better show the difference between this approach and these alternatives to make it clear.

- A minor suggestion: I believe the reachability analysis should be included in the main paper, as it is crucial to justifying the objective function and its alignment with the theoretical guarantees.

**W3**: [Empirical parts] Following FCDL, this paper evaluates the proposed method on environments including Robosuite and simulated chain and fork causal graphs. I believe these settings do demonstrate the effectiveness of the approach. However, from the perspective of the broader ambition expressed in the paper, I would encourage the authors to conduct more extensive evaluations to further strengthen the work.

- The paper frequently highlights the potential of this approach for world models. However, there is no experimental validation in this direction. In world model settings, it is common to consider an observation function, whereas the current experiments seem to focus only on low-dimensional state inputs. While this is not a major issue, it is worth acknowledging and clarifying.

- I believe it would be valuable to show the learned causal graphs in manipulation tasks with more complex structures, as done in CDL (see Fig. 22: https://proceedings.mlr.press/v162/wang22ae/wang22ae.pdf). This would help validate the method's ability to recover causal structures when more variables are involved.

- I suggest including additional baselines that infer time-varying or context-specific causal structures directly from learned model components, such as transformer attention weights or the Jacobian matrices of MLP layers. These alternatives can often be more flexible within the world model paradigm, as they do not require learning a separate latent space. Comparisons with such baselines would provide stronger empirical support for the proposed approach. In particular, works like [Pitis et al., 2020], [Urpí et al., 2024], and potentially [Lei et al., 2024] (though I understand there is no code available for this one) appear relevant and could likely be tested with minimal additional effort.

- Any empirical validation of the reachability of the intervention?

**W4**: [Discussions on related works] First, to echo my earlier point, I believe FCDL and NCD deserve more explicit acknowledgement to make the contributions of this work clearer. Additionally, you mentioned that [Pitis et al., 2020] relies on strong prior knowledge. However, after reading that paper myself, I found the approach to be quite flexible. Could you clarify specifically what strong priors are required in their method?

**W5**: [Minor] 1. The authors could consider including the corresponding graphical model and specifying additional assumptions on the graph structure (e.g., no instantaneous relationships) to make the framework more self-contained and easier to follow. 2. A typo in the expression for $\mathcal{L}{\text{quantization}}$? I suspect that in the second term $e_u$ should be ${E}_\phi(X)$ instead? Correct me if I am mistaken.

------

**References**

Hwang, Inwoo, et al. "Fine-grained causal dynamics learning with quantization for improving robustness in reinforcement learning." arXiv preprint arXiv:2406.03234 (2024).

Hwang, Inwoo, et al. "On discovery of local independence over continuous variables via neural contextual decomposition." Conference on Causal Learning and Reasoning. PMLR, 2023.

Pathak, Deepak, et al. "Curiosity-driven exploration by self-supervised prediction." International conference on machine learning. PMLR, 2017.

Kim, Kuno, et al. "Active world model learning with progress curiosity." International conference on machine learning. PMLR, 2020.

Kauvar, Isaac, et al. "Curious replay for model-based adaptation." arXiv preprint arXiv:2306.15934 (2023).

Pitis, Silviu, Elliot Creager, and Animesh Garg. "Counterfactual data augmentation using locally factored dynamics." Advances in Neural Information Processing Systems 33 (2020): 3976-3990.

Lei, Anson, Bernhard Schölkopf, and Ingmar Posner. "SPARTAN: A Sparse Transformer Learning Local Causation." arXiv preprint arXiv:2411.06890 (2024).

Urpí, Núria Armengol, et al. "Causal action influence aware counterfactual data augmentation." arXiv preprint arXiv:2405.18917 (2024).

---

> ### Author Rebuttal · Authors · 2025-07-30
>
> We thank the reviewer for acknowledging the importance of meta-causal graph (MCG), the effectiveness of our methodology, and the strength of our theoretical results on the identifiability of MCG. Below are our responses. Issues you specifically raised are marked with $\star$.
>
> > $\star$ **W1: Simlarity with FCDL/NCD**
>
> FCDL and NCD are good works, but our method is totally different.
>
> **Different Motivation**：
> - FCDL & NCD are **observation-based methods** restricting themselves to **passive learning** and limits their capacity to truly explore or adapt in an open-ended world, while our method is **intervention-based**, which helps us to deal with the open-ended world problem.
> - Our goal is to **actively discover** the unseen meta state in open-ended worlds to learn a **MCG** (line 39-40).
>
> **Different Formulation and theoretical contributions:**
> - **FCDL & NCD do not model the minimal causal graph and cannot guarantee identifiability**; their approach can lead to non-identifiable solutions due to infinite subgraph combinations. While we develop a theory framework to ensure identifiability of MCG.
> - FCDL/NCD treats actions as part of **observational data**, which does not allow for active intervention. Our method treats actions as **interventions**, enabling true active intervention for causal discovery in open worlds.
>
> > $\star$ **W2.a: Tractability of entropy computation and robustness to inaccurate posterior dist**
>
> **The computation of entropy in Eq. 2 is tractable.**
> - Entropy is calculated as the sum of the entropies of Bernoulli variables, where each variable $M[i, j]$ represents the existence of an edge $i\to j$ in the causal subgraph and the corresponding Bernoulli probabilities $\hat{M}[i, j]=P(M[i, j]=1)$ are predicted by the neural network.
> - Implementation details are provided in code file `fcdl/utils/utils.py`, function `calculate_entropy`.
>
> We **added experiments testing robustness to posterior inaccuracies** by introducing disturbances. These results show the robustness of our method.
> |Acc↑|n=2|n=4|n=6|
> |-|-|-|-|
> |FCDL|57.8|49.3|47.7|
> |**MCG (disturbed)**|**60.1**|**51.0**|**52.7**|
> |**MCG**|**63.2**|**50.5**|**50.0**|
> > $\star$ **W2.b: Discussion with other curiosity methods**
>
> Generally, curiosty-based RL methods require designing reward encouraging exploration, such as [Pathak et al., 2017], [Kim et al., 2020] and [Kauvar et al., 2023]. Compared with these methods, our curiosity-driven intervention is tailored for discovering new meta states in the MCG, which is unique to our framework, while traditional methods do not target unseen meta states.
>
> We will discuss it more in the revised version.
> > **W2.c: Move the reachability analysis from appendix to the main paper**
>
> Thank you for the suggestion. We will move the reachability analysis from appendix to the main paper in the revised version.
>
> > **W3.a: Compared with traditional world models, why not use an observation function to obtain representations?**
>
> “World model” is a broad term. Not all world models operate directly from high-dimensional observations; some, including [e.g., Liu et al.], focus on **extracting certain semantic representations** from the state space. Similarly, we **learn unknown meta state representations** and their corresponding causal graphs directly from the state space.
> > $\star$ **W3.b: Display the learned causal graphs in more complex tasks**
>
> We have **added a manipulation task with a more complex structure**. Due to space constraints, we present only a subset of the graph. This experiment validates our method’s ability to recover causal structures in settings with 15 variables. Larger one will be added in next version.
>
> **MCG**
> ||obj 0|obj 1|obj 2|obj 3|obj 4|
> |-|-|-|-|-|-|
> |obj 0|1.0|0.0|0.0|0.0|0.0|
> |obj 11|1.0|1.0|0.5|0.5|0.6|
> |obj 12|0.6|0.7|0.9|0.0|0.1|
> |obj 13|1.0|0.9|0.7|0.0|0.4|
>
> **Ground truth**
>
> ||obj 0|obj 1|obj 2|obj 3|obj 4|
> |-|-|-|-|-|-|
> |obj 10|1.0|0.0|0.0|0.0|0.0|
> |obj 11|1.0|1.0|1.0 |1.0|1.0|
> |obj 12|1.0|1.0|1.0|0.0|0.0|
> |obj 13|1.0|1.0|1.0|0.0|0.0|
>
> > $\star$ **W3.c: Add additional baselines**
>
> We **added baselines you suggested**: causal discovery using transformer attention weights ("transformer") and the Jacobian matrices of MLP layers ("sandy-mixture") from [Pitis et al., 2020] and [Urpí et al., 2024], for comparison. The results are shown below. Our method outperforms new baselines, demonstrating that our active intervention strategy improves the accuracy of the recovered causal graph.
>
> |Acc↑|Fork (n=2)|Fork (n=4)|Fork (n=6)|Chain (n=2)|Chain (n=4)|Chain (n=6)|
> |-|-|-|-|-|-|-|
> |transformer|24.45|22.49|21.41|29.62|30.37|29.78|
> |sandy-mixure|31.93|32.47|33.72|30.08|29.31|27.43|
> |**MCG**|**63.18**|**50.47**|**50.04**|**51.99**|**49.78**|**49.69**|
>
> |Acc↑|Fork (n=2)|Fork (n=4)|Fork (n=6)|Chain (n=2)|Chain (n=4)|Chain (n=6)|
> |-|-|-|-|-|-|-|
> |transformer|6.20|6.13|6.71|6.67|6.73|7.40|
> |sandy-mixure|6.48|6.10|6.50|7.03|6.73|7.17|
> |**MCG**|**14.65**|**14.06**|**13.28**|**13.82**|**12.49**|**12.45**|
> > $\star$ **W3.d: Empirical validation of reachability**
>
> We **added experiments comparing causal graphs under limited vs. full reachability**. Limited reachability yields less accurate graphs (higher SHD, lower F1).
> ||SHD↓|F1↑|
> |-|-|-|
> |MCG (limited reachability)|17.31|0.77|
> |MCG|11.69|0.84|
>
> > **W4.a: Discussions on related works**
>
> We acknowledge the valuable contributions and inspiration of FCDL/NCD in causal discovery and will add more discussion in the revision. We believe that key conceptual differences have already been clarified in our response to **Comment W1**. We provide other comparisons here:
> - Observation-based methods like FCDL and NCD are limited to Markov equivalence classes, while intervention-based methods (Ours) can **actively reduce the Markov equivalence class and uniquely identify causal relations** that are otherwise unidentifiable.
> - We introduce **$L_{mask}$ loss** (line 197-198) that directly updates causal structure based on intervention outcomes while FCDL & NCD have no mechanism to verify or update causal beliefs through interventions.
>
> > **W4.b: Citation for [Pitis et al., 2020]**
>
> Apologies for the citation error. The intended reference was another paper, [Pitis et al., 2022], which discusses methods that require domain knowledge or expert trajectories. We will correct this mistake in the revised version.
>
> > **W5: Typos and clarification of theoretical aspects of the paper**
>
> Thank you for your careful reading. We will improve the clarity of the theoretical aspects of the paper and fix the typos.
>
> > **Q1.a: How to integrate and learn context-specific causal discovery approach in world model efficiently**
> - Our approach uses a curiosity-driven exploration strategy to **actively discover unknown meta states**. Specifically, the agent **maps each state to a learned representation** that jointly captures the meta state and its corresponding causal graph.
> - Based on the discovered causal graphs, we learn a transition model conditioned on meta states. Here, the **MCG** serves as **structured world knowledge**, while the **learned transition model functions** as the **world model** (see Definition 3 and line 204-205).
> - We added experiments to show efficiency of our method. We compare other methods (GNN and MLP) with longer training rounds and more parameters. The result is shown below:
>
> **Comparison with _Same_ Model Size**
>
> ||Acc n=2|Acc n=4|Acc n=6|Reward n=2|Reward n=4|Reward n=6|
> |-|-|-|-|-|-|-|
> |GNN (30k round)|38.13|25.81|20.75|6.83|6.06|6.83|
> |GNN (25k round)|36.37|26.55|20.73|6.84|5.98|6.19|
> |MLP (30k round)|30.47|28.94|27.84|6.20|6.13|6.71|
> |MLP (25k round)|31.47|30.32|30.42|6.18|6.03|6.23|
> |**MCG** (10k round)|**47.01**|**48.37**|**46.35**|**11.96**|**10.83**|**10.70**|
> |**MCG** (12k round)|**48.30**|**48.40**|**47.28**|**12.45**|**10.77**|**10.84**|
> |**MCG** (15k round)|**71.56**|**59.17**|**49.58**|**14.65**|**14.06**|**13.28**|
>
> **Comparison with _Same_ Round**
> ||Acc n=2|Acc n=4|Acc n=6|Reward n=2|Reward n=4|Reward n=6|
> |-|-|-|-|-|-|-|
> |MLP (parameter $\times$ 3.71)|24.65|25.05|23.71|6.29|6.50|7.93|
> |GNN (parameter $\times$ 1.66)|37.58|30.61|30.05|6.34|6.49|7.90|
> |**MCG** (parameter $\times$ 1)|**71.56**|**59.17**|**49.58**|**14.65**|**14.06**|**13.28**|
>
> Our method achieves strong performance with less training data and smaller models, highlighting its efficiency.
> > **Q1.b: How should actions be incorporated into the framework?**
>
> In our framework, actions are treated as **interventions** for causal discovery.
>
> > **Q1.c: Additional structural assumptions needed?**
>
> While not all actions can intervene on every state (i.e., not all states are reachable by intervention), we explicitly make a **reachability assumption** to ensure identifiability of the underlying structure.
>
> > **Q1.d: Can you leverage this structure discovery as part of reward learning?**
>
> Our curiosity-driven reward (Eq. 2) is explicitly designed to encourage MCG structure discovery of meta-states $u=C(x)$ and subgraphs $\mathcal{G}_u$.
>
> > **Q2.a: Clarification of observation equivalence**
>
> Thank you for your comment. We believe there may be a slight misunderstanding. Observational equivalence refers to the condition where two mappings from states to meta states produce the same partitioning of observations, possibly up to relabeling. We will clarify this in the revised version.
>
> > **Q2.b: How can imperfect representation affect the identifiability of meta-states and the causal graph?**
>
> Thank you for your comment. We agree that learned latents can be imperfect. Empirically and theoretically, our study on (Fig. 2 and Thm. 2) shows over‑parameterisation does not hurt performance much while under-parameterisation will lead to degradation of performance. We will add more discussions on this point in the revised version.
>
> Yang Liu, et al. Learning world models with identifiable factorization. NeurIPS, 2023.
>
> Silviu Pitis, et al. Mocoda: Model-based counterfactual data augmentation. NeurIPS, 2022.

---

> ### Comment · Reviewer_hzVP · 2025-08-01
>
> Many thanks for the detailed response. I really appreciate the new experimental results and clarifications. That said I still have several concerns and would be glad to continue the discussion.
>
> On W1: Yes, this is precisely what I meant in my original review: the core novelty lies in the added “active intervention” module based on entropy, while the rest of the pipeline remains quite similar to prior work. I understand the point of identifiability, but from a technical standpoint, the contribution still feels incremental. I would suggest spending more effort in future versions to explicitly position the work relative to prior methods and better highlight what is new.
>
> On W2-a: Could you clarify how posterior accuracy was computed in the new experiment, was it over the full graph or limited to specific subgraphs?
>
> On W2-b: I am not fully convinced by this. The cited methods are indeed reward-free, and the intrinsic signals they use (e.g., based on state properties such as uncertainty) serve similar purposes. In your case, entropy-based reward is a well-established form of intrinsic motivation. If the claim is about discovering novel meta-state representations, many of those methods can arguably do the same under different selection criteria. Happy to continue the discussion here.
>
> On W3-a: I understand your argument, but I just checked [Liu et al.]—they do learn the world model from pixels. You may want to reconsider this citation. Also, please double-check your citation, looks like the author name may be incorrect
>
> On W3-b: Thanks for the new results. It would be much better to also include a baseline for comparison, and report standard deviations (this setting can be sensitive to random seeds)
>
> On Q1-b: Framing all actions in a world model as interventions feels too strong. In many real-world  tasks, most actions do not intervene in a meaningful way (e.g., small or redundant movements). IMO, this assumption weakens the world model framing
>
> On Q2-b: My point was not just about under- or over-parameterization, but more generally about lossy representations. This can arise from multiple sources. It would be wonderful to give more theoretical analysis on this point
>
> Overall, I sincerely thank the authors for the thoughtful and thorough response. While I understand the framing around active interventions, I remain unconvinced about positioning this as a “world model” paper due to the concerns above. Also, the method feels incremental and lacks sufficient acknowledgement of related baselines (At the very least, I would suggest that in the next version, you explicitly state how your method builds on others, as the current presentation may give readers a misleading impression). Also, the entropy-based criteria for intervention are not entirely new. Framing it as a causal discovery work may be more appropriate. In that case, I would encourage more thorough discussion and comparison to active causal discovery and experimental design literature (I listed a few below).
>
> I would look forward to further conversations and appreciate the effort and time the authors have put into the response.
>
>
> Choo, Davin, Themistoklis Gouleakis, and Arnab Bhattacharyya. "Active causal structure learning with advice." International Conference on Machine Learning. PMLR, 2023.
>
> Zhang, Zeyu, et al. "Bayesian active causal discovery with multi-fidelity experiments." Advances in Neural Information Processing Systems 36 (2023): 63957-63993.
>
> Zhang, Jiaqi, et al. "Active learning for optimal intervention design in causal models." Nature Machine Intelligence 5.10 (2023): 1066-1075.
>
> Elahi, Muhammad Qasim, et al. "Adaptive online experimental design for causal discovery." arXiv preprint arXiv:2405.11548 (2024).
>
> Huang, Daolang, et al. "Amortized bayesian experimental design for decision-making." Advances in Neural Information Processing Systems 37 (2024): 109460-109486.

---

> ### Author Response · Authors · 2025-08-04
> **Core Contributions and Relation to Prior Work**
>
> We sincerely thank the reviewer for carefully reading our paper and rebuttal, and for expressing interest in further discussion. We truly appreciate the thoughtful engagement, and we also hope this exchange can lead to a deeper and more meaningful dialogue.
>
> > **(Re: W1 W3-b) : Discussion on FCDL, Our Contributions, and Relation to Active Causal Methods** ($\star$)
>
> * We acknowledge that the primary algorithmic novelty of our work lies in the entropy-based curiosity feedback. However, the core contribution of our paper is theoretical: we are motivated by the challenge of causal dynamics in open-ended worlds, and we address whether it is possible to identify context-dependent causal graphs (meta-causal graphs) through interventions.
>
>     In our previous response, due to limited space, we mainly highlighted the differences with FCDL. In fact, only in Section 4.1 (Learning from Experiences), we referenced and built upon FCDL. Therefore, it is understandable that the reviewer sees our method as an extension of FCDL. **However, our main contribution goes beyond FCDL.** We provide a theoretical framework for identifying context-dependent causal graphs under interventions (Section 4), and design a curiosity-driven agent to actively explore and distinguish meta states and their associated subgraphs. The key challenge is how to detect meta states and how to learn different causal structures conditioned on them continuously in open-ended world. We will clarify this in the revised version.
>
> * The active interventions in our method connects our work to the literature on active causal discovery and experimental design. With reference to the papers suggested by the reviewers, we outline the key differences below.
>     [1, 2, 4]: These works focus on active exploration to discover causal relations, but they **do not model context-specific causal graphs**. They assume that causal discovery accumulates into a single static graph as more data is observed. In contrast, we e**xplicitly model context-dependence**, where the agent must jointly learn the meta state and its corresponding subgraph. This requires a new theoretical treatment, which we provide.
>
>     [3, 5]: These works propose intervention strategies, but **their goal is not causal discovery**. Instead, they aim to design effective interventions to facilitate scientific experimentation. In contrast, our method aims to **learn the causal structure itself under context shifts through interaction**.
>
> To make it more clear, we put detailed discussion of our motivation and discussion with context-dependent like (FCDL) at the end of reply.
>
> [1] Choo, Davin, Themistoklis Gouleakis, and Arnab Bhattacharyya. "Active causal structure learning with advice." International Conference on Machine Learning. PMLR, 2023.
>
> [2] Zhang, Zeyu, et al. "Bayesian active causal discovery with multi-fidelity experiments." Advances in Neural Information Processing Systems 36 (2023): 63957-63993.
>
> [3] Zhang, Jiaqi, et al. "Active learning for optimal intervention design in causal models." Nature Machine Intelligence 5.10 (2023): 1066-1075.
>
> [4] Elahi, Muhammad Qasim, et al. "Adaptive online experimental design for causal discovery." arXiv preprint arXiv:2405.11548 (2024).
>
> [5] Huang, Daolang, et al. "Amortized bayesian experimental design for decision-making." Advances in Neural Information Processing Systems 37 (2024): 109460-109486.
>
> > **(Re: On W2-a): Could you clarify how posterior accuracy was computed in the new experiment, was it over the full graph or limited to specific subgraphs?**
>
> We added noise to the input to obtain an inaccurate posterior. It was over full graph.
>
> > **(Re: On W2-b) On the Design of Entropy-Based Objective**
>
> * We would like to clarify that the use of entropy in our method serves primarily as a **carrier of the exploration objective**.
> * The key idea lies in the construction of the meta-causal graph embedded inside the entropy calculation.
> * In fact, **entropy in our method could be replaced by other measures of uncertainty**. The role of the intrinsic reward is to encourage the agent to explore previously unseen meta states.
> * Each meta state is tightly associated with a distinct causal graph, so this exploration strategy effectively pushes the agent toward regions where causal structures are most uncertain or subject to change.
> * While entropy-based exploration has been used in prior work, entropy also functions there as a proxy. **What really matters is the goal encoded inside the entropy term**. In our case, it is to identify causal changes across contexts.

---

> > ### Comment · Reviewer_hzVP · 2025-08-05
> > **Response to "Core Contributions and Relation to Prior Work"**
> >
> > Thank you for the detailed response. I will list my comments for each of the points below.
> >
> > >(Re: W1, W3-b)
> > Yes, I understand the motivation here. However, my concern remains regarding the writing, which could be more clearly structured in future versions. Specifically, it would help to formally distinguish which components are based on prior work and which technical parts are novel.
> >
> > >(Re: W2-a)
> > Got it. Thank you for the explanation.
> >
> > >(Re: W2-b)
> > Yes, this is precisely what I was trying to express in my previous feedback. It would be valuable to consider other exploration strategies as well. While I understand the rationale behind using entropy, since the goal is to propose a general framework, it would be ideal if it could accommodate other intrinsic terms that share high-level similarities with entropy.

---

> ### Author Response · Authors · 2025-08-04
> **Definition and Scope of World Models and Additional Experiments**
>
> > **(Re: W3-a) Why this method can be regarded as a world models method?**($\star$)
>
> * **The definition of world models remains an open topic of discussion in the literature**, generally falling into two major perspectives: (i) **understanding the world**, and (ii) **predicting the future** [7], enabling AI agents to plan, reason, and make decisions.
> * From a broad perspective, we consider world models is to **include the state and a transition function from (state, action) to next state**. Thus, world models are not limited to model-based RL frameworks with an added observation function. Rather, any approach that captures the regularities and dynamics of the world can be referred to as a world model [6].
> * This broader interpretation in causality is also supported by Bernhard Schölkopf’s keynote at ICDM, Towards Causal World Models and Digital Twins, where he argues that **causal models represent a form of world modeling**. In particular, the transition function from (state, action) to next state can be derived from a structural causal model.
> * Based on this view, methods that learn world dynamics, including FCDL, can be considered world models.
>
> Under a narrower interpretation of world models as "learning from observations", we would like to highlight that
> 1. our method learns a **meta state representation**, which constitutes a mapping from high-dimensional state space to a lower-dimensional meta state space,
> 2. based on the meta-causal graph, we further learn a **transition models** show interaction dynamics of transition model, they are updated continuously by active agent exploration. This aligns with the goal of learning a compact, predictive model of the world.
>
> [6] Marvin Minsky. A framework for representing knowledge, 1974.
>
> [7] Jingtao Ding et al. Understanding World or Predicting Future? A Comprehensive Survey of World Models
>
> > **(Re: On W3-b) It would be much better to also include a baseline for comparison, and report standard deviations (this setting can be sensitive to random seeds)**
>
> Thank you for your advice. We **added more experiments** to report standard deviations.
>
> ||obj 0|obj 1|obj 2|obj 3|obj 4|
> |-|-|-|-|-|-|
> |obj 10|0.56 $\pm$ 0.21|0.58 $\pm$ 0.20|0.57 $\pm$ 0.19|0.33 $\pm$ 0.29|0.44 $\pm$ 0.25|
> |obj 11|0.60 $\pm$ 0.19|0.56 $\pm$ 0.21|0.57 $\pm$ 0.20|0.32 $\pm$ 0.28|0.42 $\pm$ 0.26|
> |obj 12|0.58 $\pm$ 0.21|0.58 $\pm$ 0.19|0.57 $\pm$ 0.19|0.34 $\pm$ 0.29|0.40 $\pm$ 0.29|
> |obj 13|0.60 $\pm$ 0.20|0.60 $\pm$ 0.21|0.57 $\pm$ 0.18|0.36 $\pm$ 0.29|0.37 $\pm$ 0.27|
>
> ||obj 0|obj 1|obj 2|obj 3|obj 4|
> |-|-|-|-|-|-|
> |obj 10|0.45 $\pm$ 0.24|0.45 $\pm$ 0.20|0.42 $\pm$ 0.21|0.42 $\pm$ 0.21|0.50 $\pm$ 0.24|
> |obj 11|0.50 $\pm$ 0.21|0.42 $\pm$ 0.24|0.42 $\pm$ 0.19|0.35 $\pm$ 0.20|0.58 $\pm$ 0.21|
> |obj 12|0.49 $\pm$ 0.20|0.38 $\pm$ 0.18|0.46 $\pm$ 0.19|0.38 $\pm$ 0.18|0.57 $\pm$ 0.24|
> |obj 14|0.45 $\pm$ 0.18|0.52 $\pm$ 0.20|0.42 $\pm$ 0.16|0.36 $\pm$ 0.23|0.46 $\pm$ 0.25|
>
> We **added new experiment** to evaluate SHD and F1 compared with FCDL. The results shows that causal graph recovered by MCG is more accurate.
> ||SHD|F1|
> |-|-|-|
> |FCDL|30.00 $\pm$ 7.16|0.52 $\pm$ 0.05|
> |**MCG**|**11.00 $\pm$ 7.10**|**0.81 $\pm$ 0.06**|
>
> > (Re: On Q1-b) how to deal with if actions do not intervene in a meaningful way?
>
> * We have considered this issue and discuss it explicitly in the paper.
> * Specifically, we provide a theoretical discussion under the Reachability assumption, detailed in Theorem 3 (Appendix, Lines 671–697).
> * The theorem shows that as long as a state can be directly or indirectly influenced by actions, the corresponding meta-causal graph is identifiable.
> * We also acknowledge this as a limitation in our Limitations section, where we state that our current method assumes all states are reachable, either through direct or indirect interventions.
> * We **added a new experiment** based on a manipulation task to demonstrate that our method can handle cases **where actions cannot intervene on all variables**. We also modified this environment to test generalization. The results show that our method maintains strong performance and generalizes well, even when only a subset of variables is directly intervenable.
>
> |Reward↑|Small Magnetic Force ($\times 0.02$)|High Ball Density ($\times$ 10)|Extra Table Friction|
> |-|-|-|-|
> |MLP|4.011 ± 0.030|3.999 ± 0.038|3.886 ± 0.241|
> |FCDL|4.482 ± 0.622|4.451 ± 0.657|4.193 ± 0.655|
> |**MCG**|**6.173 ± 0.255**|**5.083 ± 0.245**|**5.517 ± 0.134**|

---

> > ### Comment · Reviewer_hzVP · 2025-08-05
> > **Response to "Definition and Scope of World Models and Additional Experiments"**
> >
> > >(Re: W3-a)
> > I understand this point. What I mentioned previously was indeed an incorrect citation to support your claim. However, regarding the framing of the world model, I believe the role of actions should be taken into account, which I assume will be discussed later in the paper.
> >
> > >(Re: W3-b)
> > This is now clear, thank you!
> >
> > >(Re: Q1-b)
> > I appreciate your response and the accompanying evaluations. My main point was that not all actions should be treated as interventions. I just wanted to flag this and suggest that a more formal definition could be helpful in future versions. That said, I see that you've acknowledged this point here, which is good.

---

> ### Author Response · Authors · 2025-08-04
> **Theoretical Analysis of Lossy Representations**
>
> > **(Re: On Q2-b) Theoretical analysis of lossy representations**
>
> | Symbol                 | Meaning                             |
> | ---------------------- | ----------------------------------- |
> | $U={u}$              | ground-truth meta states            |
> | $\hat U={\hat u_k}$ | prototype (codebook) embeddings     |
> | $C(x)$               | true meta state of sample $x$          |
> | $\hat C(x)$          | prototype selected by encoder $E$ |
> |$\mu_u$| prior probability of state $x$ with true meta state $u$|
> |$p_k = \sum_{u\in U} \mu_up_k^u$| probability of mapping sample $x$ to prototype embeddings $e_{\hat{u}_k}$|
>
> The lossy representation affects the probability of mapping a sample $x$ to the prototype embeddings.
>
> We assume that for each state $x$ with true meta state $C(x) = u$, the probability that it is mapped by the encoder $E$ to the prototype embedding $\hat{u}_k$ is $p_k^u = P(\hat{C}(x) = \hat{u}_k \mid C(x) = u)$. **This probability is determined by the representation power and discriminability of the encoder $E$**, and it reflects the encoder’s tendency to assign samples with the same true meta state $u$ to different prototype embeddings.
>
> From Definition 5, $\hat{C}(x)$ and $C(x)$ are observationally equivalent if, for all $\hat{u}_k \in \hat{U}$ and for all $x_1, x_2 \in \\{x \mid \hat{C}(x) = \hat{u}_k\\}$, $C(x_1) = C(x_2)$.
>
> $$
> P(C(x) = u, \forall x' \in \{x \mid \hat{C}(x) = \hat{u}_k\}, C(x') = u) = (1 - p_k + p_k^u \mu_u)^n,
> $$
>
> where $n$ is the number of samples.
>
> The probability of $\hat{C}(x)$ and $C(x)$ are observationally equivalent  is given by $\sum_{k} p_k \sum_{u \in U} \mu_u (1 - p_k + p_k^u \mu_u)^n.$
>
> For a sample $x$, the probability that, for all $x' \in \\{x' \mid \hat{C}(x') = \hat{C}(x)\\}$, $C(x') = u' \neq C(x) = u$ is:
>
> $$
> P(\forall x' \in \{x' \mid \hat{C}(x') = \hat{C}(x)\}, C(x') = u' \neq C(x) = u) = \sum_{k}\sum_{u \in U}(p_k - \mu_u p_k^u)(1 - p_k + p_k^u \mu_u)^n.
> $$
>
> For a sample $x$, the probability of $x$ being misclassified as belonging to a representation entry $e_u$ that has multiple true meta states is:
>
> $$
> P(\exists x_1, x_2 \in \{x' \mid \hat{C}(x') = \hat{C}(x)\}, C(x_1) \neq C(x_2)) = 1 - \sum_{k} p_k \sum_{u \in U} (1 - p_k + p_k^u \mu_u)^n.
> $$
>
> Due to the lossy representation, the probability of misclassification is:
>
> $$
> \begin{align*}
> P(\text{misclassification}) &= 1 - \sum_{k}\sum_{u \in U} \mu_u p_k^u (1 - p_k + p_k^u \mu_u)^n \\
> \end{align*}
> $$
>
> If $n(p_k + p_k^u \mu_u) \ll 1$, for representation entry $e_u$,
>
> $$
> P(\text{misclassification}) \approx n \sum_{u}\sum_k\mu_u p_k^u (p_k - p_k^u \mu_u)=\sum_k[p_k^2-\sum_u (p_k^u \mu_u)^2]
> $$
>
> In a more accurate representation, for each prototype $k$ the vector $\{\mu_u p_k^u\}_u$ concentrates on a single true state—one $p_k^u$ increases while the others decrease—thereby increasing $\sum_u (\mu_u p_k^u)^2$ and thus reducing each bracketed term $\bigl[p_k^{2}-\sum_u (\mu_u p_k^u)^2\bigr]$.
> Consequently, higher representation accuracy monotonically lowers the error bound, reaching zero contribution per prototype when it is class-pure.

---

> > ### Author Response · Authors · 2025-08-04
> > **Motivation and Comparison with Context-Dependent Causal Discovery**
> >
> > > **More discussion about our motivation and comparison with context-dependent causal discovery**
> > 1.	On Motivation and Comparison to context-dependent causality ([8], chapter 10 [10], FCDL)
> >     * The initial motivation for our work came from a fundamental question: **Why do causal discovery methods often fail in open-ended environments?**
> >     * We observed that causal models aim to describe the rules of the world. However, in open-ended settings (e.g., open-ended games, multi-agent systems, or LLM-based dialogue), no single observation window can capture all possible events. This leads to instability: a causal relation valid in one context may fail in another.
> >     * For example, Newtonian physics holds at the macroscopic level but breaks down at the quantum scale. FCDL also emphasizes this issue (but it’s not the first one), arguing that **a complete causal law must account for variation across different contexts**. Therefore, our high-level motivation aligns with them. many previous works have explored this motivation context-dependent causal structures, as we briefly discussed in Lines 226–230.
> >     * However, FCDL is not the first one discusses about context-dependent causality. The idea of context-dependent causality was formally introduced by earlier Huang et al. [8], and also discussed earliest in Chapter 10 of Pearl’s Causality [9].
> > 2.	On Exploration and the Limitation of Passive Learning (([8], FCDL))
> >     * A key challenge in scaling up to open-ended worlds is how to explore effectively. Relying solely on random exploration or limited observed data is insufficient to discover hidden causal mechanisms.
> >     * For instance, in scientific discovery, new causal knowledge often emerges only after targeted investigation of paradoxes. Similarly, in dialogue agents, causal relationships may even reverse depending on the context.
> >     * Unlike prior works including FCDL and [8], which rely solely on observation passive data, our method explicitly considers **the unobserved causal space**, and **how newly explored data may lead to shifts in both the causal graph and the context boundary**.
> > 3. On Contribution Beyond FCDL: Causality-Seeking Agent
> >     * Our title, Causality-Seeking Agents, reflects our core contribution: **going beyond passive modeling to active exploration for causal discovery**.
> >     * The reviewer commented that our curiosity and intervention-based strategy is incremental. But this form of self-improving agent behavior addresses a core limitation of data-driven methods: **the inability to discover unknown causal structures**.
> >     * However, purely exploring from scratch is inefficient. Without prior knowledge, agents are essentially blind.
> >     * Therefore, we include a learning-from-experience phase (Section 4.1), implemented within the FCDL environment. This provides a strong initialization, similar in spirit to pre-training in large models.
> >     * The main focus of our method lies in Section 4.2 and 4.3, which describe a progressive learning process through active intervention.
> >     * Our primary theoretical contribution is **the construction of a formal intervention-based meta-causal graph, with associated theorems and learning framework**.
> > We will include all the discussion and emphasis the importance of FCDL to our next version as the base of context-aware causality method.
> >
> > [8] Biwei Huang et al.. Causal discovery from heterogeneous/nonstationary data. Journal of Machine Learning Research
> >
> > [9] Judea Pearl, Causality

---

> > ### Comment · Reviewer_hzVP · 2025-08-05
> > **Response to "Theoretical Analysis of Lossy Representations" & "Motivation and Comparison with Context-Dependent Causal Discovery"**
> >
> > For these two points, I will need a bit more time to review and will likely get back to you once I finish (likely by the 6th). I don’t want to cause unnecessary delay, but I would like to make a fair assessment. Thank you so much for your patience!

---

> > > ### Author Response · Authors · 2025-08-05
> > >
> > > Dear Reviewer hzVP,
> > >
> > > We sincerely appreciate your commitment to evaluating our paper despite your busy schedules. Thank you for your encouraging words and valuable feedback!
> > >
> > > We are also working on more settings and will update you once the results are available.
> > >
> > > Many thanks,
> > >
> > > The authors of #20694

---

> > > ### Comment · Reviewer_hzVP · 2025-08-07
> > > **Re "Motivation and Comparison with Context-Dependent Causal Discovery"**
> > >
> > > 1. Regarding the term "open-ended tasks", such as open-ended games, multi-agent systems, or LLM-based dialogue. I feel this may be somewhat of an overstatement. While it's a good motivational point, the current framework does not actually demonstrate benefits in these settings. I would suggest avoiding this claim.
> > >
> > > 2. On the limitation of passive learning: I completely understand your point. However, I would consider framing it more as a distinction between "observation-based causal discovery" and "intervention-based" approaches. What do you think?
> > >
> > > 3. As for the contribution, I understood it from the beginning of the review process, but I really appreciate your explanation here.

---

> > > > ### Author Response · Authors · 2025-08-07
> > > >
> > > > > Regarding the term "open-ended tasks", such as open-ended games, multi-agent systems, or LLM-based dialogue. I feel this may be somewhat of an overstatement. While it's a good motivational point, the current framework does not actually demonstrate benefits in these settings. I would suggest avoiding this claim.
> > > >
> > > > Thank you very much for your thoughtful feedback and for raising this important point.
> > > >
> > > > * We share the same perspective that in some open-ended tasks, such as multi-agent environments, video games, LLM-based dialogue, are highly challenging and require long-term efforts. While our experiments do not yet cover all chanllenge large scale open-endedness scenarios, **our paper are motivated by three core challenges that commonly arise in such settings**:
> > > >
> > > >   - a. causal relationship is context-dependent.
> > > >   - b. hard to rely only on statistical learning from observation like independent test (FCDL and Huang [1]).
> > > >   - c. lack of efficient exploration to discover unknown changes.
> > > >
> > > >   We also provide a theoretical foundation that discusses identifiability boundaries under these conditions, which we believe lays a useful groundwork for tackling broader open-ended problems.
> > > >
> > > >
> > > > * To address your concern, we will expand the discussion on **potential extensions of our method to more open-ended settings**. For example, regarding potential applications to LLM-based dialogue in wild, in such scenario the main open-endlessness chanlleges is that it often difficult to collect enough event pairs samples to perform statistical causal discovery. Instead, intervention-based discovery and efficient exploratory generation become necessary. One potential solution we will discuss is to deploy a curiosity-driven chat agent that interacts with the LLM through curious-based targeted, causal-oriented queries, gradually constructing a causal graph of its internal behavior. While this involves a different experimental design, it is fully aligned with the principles of our framework and theorem, context-awareness and curiosity-driven intervention.
> > > >
> > > > * To further support our claims, we **have added a new experiment** that explicitly simulates challenges arising from limited observation windows, aligning with our original motivation (Introduction, lines 32–43). This setting evaluates how our method performs when new variables gradually become observable. The results demonstrate that our method is robust under partial observability and holds promise for more realistic, evolving environments:
> > > >
> > > > |Prediction Accuracy|n=2|n=4|n=6|
> > > > |-|-|-|-|
> > > > |MLP|31.93 $\pm$ 0.16|32.47 $\pm$ 1.61|33.72 $\pm$ 2.11|
> > > > |FCDL|63.75 $\pm$ 13.75|53.89 $\pm$ 11.95|52.11 $\pm$ 16.43|
> > > > |**MCG**|**73.52 $\pm$ 9.56**|**63.71 $\pm$ 5.79**|**57.05 $\pm$ 14.63**|
> > > >
> > > > [1] Biwei Huang et al.. Causal discovery from heterogeneous/nonstationary data. Journal of 361 Machine Learning Research
> > > >
> > > > > On the limitation of passive learning: I completely understand your point. However, I would consider framing it more as a distinction between "observation-based causal discovery" and "intervention-based" approaches. What do you think?
> > > >
> > > > Thank you very much for the insightful suggestion.
> > > >
> > > > * We will **adopt your framing** and improve the Related Work section to better **clarify our differences compared with both "observation-based" and "intervention-based approaches"**, as discussed in our previous response.
> > > >
> > > > * Additionally, we will emphasize that our contribution goes one step further: rather than passively applying interventions, our method conducts **active exploration**, selecting interventions that are specifically aimed at uncovering and disambiguating context-dependent causal structures (meta-graphs) that may vary across different states or tasks.

---

> > ### Comment · Reviewer_hzVP · 2025-08-07
> > **Re "loss representation"**
> >
> > I don't have any particular questions about this part. Thank you for providing the details.

---

> ### Author Response · Authors · 2025-08-06
>
> > (Re: W1, W3-b) Yes, I understand the motivation here. However, my concern remains regarding the writing, which could be more clearly structured in future versions. Specifically, it would help to formally distinguish which components are based on prior work and which technical parts are novel.
>
> Thank you for the helpful suggestion. We will restructure the paper to clearly separate prior foundations from novel parts as we mentioned in *Motivation and Comparison with Context-Dependent Causal Discovery*.
> * We built the learning-from-experience framework on top of prior work, which provides a strong initialization.
> * Our novel contributions include:
>   * The construction of a formal intervention-based meta-causal graph, with associated theorems and learning framework.
>   * A progressive learning process through active intervention.
>
> > (Re: W2-b) Yes, this is precisely what I was trying to express in my previous feedback. It would be valuable to consider other exploration strategies as well. While I understand the rationale behind using entropy, since the goal is to propose a general framework, it would be ideal if it could accommodate other intrinsic terms that share high-level similarities with entropy.
>
> Thank you. We **added new experiments** with different intrinsic terms.
>
> |Prediction Accuracy|Fork (n=2)|Fork (n=4)|Fork (n=6)|Chain (n=2)|Chain (n=4)|Chain (n=6)|
> |-|-|-|-|-|-|-|
> |Prediction Uncertainty|52.25 $\pm$ 16.73|59.46 $\pm$ 22.38|52.25 $\pm$ 28.46|59.52 $\pm$ 8.82|54.81 $\pm$ 5.47|59.28 $\pm$ 5.03|
> |Feature Discrepancy|46.11 $\pm$ 8.38|41.17 $\pm$ 5.67|43.67 $\pm$ 2.47|56.01 $\pm$ 13.13|51.88 $\pm$ 7.22|58.27 $\pm$ 8.53|
> |Predictive-Distribution Discrepancy|54.23 $\pm$ 8.72|46.27 $\pm$ 9.74|43.92 $\pm$ 1.85|36.83 $\pm$ 3.11|40.33 $\pm$ 4.89|45.40 $\pm$ 2.83|
>
> We evaluated three **intrinsic objectives**:
>
> * **Prediction Uncertainty.**
>   $r_{\text{int}} = H\left[p_{\theta}(s_{t+1}\mid s_t,a_t)\right]$, i.e., the entropy of the model’s predictive distribution over the next state (or its feature representation).
>
> * **Feature Discrepancy.**
>   Let $z_{t+1}=f_{\phi}(s_{t+1})$ denote the feature representation of the next state, and let $\hat{s}\sim p_{\theta}(\cdot\mid s_t,a_t)$ with $\hat{z}=f_{\phi}(\hat{s})$. We set $r_{\text{int}}=\|\hat{z} - z_{t+1}\|_1$.
> * **Predictive-Distribution Discrepancy.**
>   $r_{\text{int}} = -\log p_{\theta}(s_{t+1}\mid s_t,a_t)$ (surprisal / negative log-likelihood). *(For discrete $s_{t+1}$, one may alternatively use $1 - p_{\theta}(s_{t+1}\mid s_t,a_t)$, but we adopt the former for scale invariance.)*
>
>
> In all cases, $r_{\text{int}}$ serves as the curiosity signal guiding actions-as-interventions in the open-ended world.
>
>
> > (Re: W3-a) I understand this point. What I mentioned previously was indeed an incorrect citation to support your claim. However, regarding the framing of the world model, I believe the role of actions should be taken into account, which I assume will be discussed later in the paper.
>
> Thank you for your comment. We will clarify the role of actions in the world model in the next revision.
>
> * Sorry about the incorrect citation. We also note that Richens and Everitt operates directly on features rather than introducing an explicit high-dimensional observation function [1]; those features are treated as directly/partially observable and (in many cases) directly intervenable, which explains their action–intervention alignment.
> * Our method identifies context-dependent causal graphs (meta-graphs) via interventions. **Actions** are how the curiosity-driven agent **realizes those interventions**.
> * When actions can **directly intervene each state** (e.g., a chemical environment), we deploy **targeted interventions** to isolate edges and quickly distinguish meta states and their subgraphs.
> * When not all interventions are feasible, we rely on the **reachability assumption** (Appendix A.8).
> * We also added an experiment comparing causal graphs under limited vs. full reachability (W3.d).
> * We will incorporate these clarifications in the next revision.
>
> [1] Richens, J., & Everitt, T. (2024). Robust agents learn causal world models. arXiv preprint arXiv:2402.10877.

---

> > ### Comment · Reviewer_hzVP · 2025-08-07
> >
> > Thank you for the additional clarifications and the new experiments on different intrinsic objectives, this makes much more sense to me now. Truly appreciate your time and effort.
> >
> > While we may have had some disagreements on certain points, I believe we share the goal of improving the quality of the work and refining the framing of its concepts and claims. If anything in the process made you uncomfortable or caused any frustration, I sincerely apologize.
> >
> > Thanks again, and I’ll proceed to update my rating.

---

> > > ### Author Response · Authors · 2025-08-07
> > > **Thank you very much for your time and thoughtful engagement!**
> > >
> > > Dear reviewer hzVP,
> > >
> > > We sincerely appreciate your recognition of our work, your thoughtful engagement, and the time you have dedicated to reviewing our paper. Your valuable suggestions have undoubtedly helped improve the quality of the work, and the discussion has been deeply thought-provoking, inspiring us to reflect more thoroughly on both our current approach and future directions. Thank you again for your interest in our research.
> > >
> > > Many thanks,
> > >
> > > The authors of #20694

---

### Official Review · Reviewer_KoPL · 2025-07-01

**Clarity:** 2
**Significance:** 2
**Originality:** 2
**Rating:** 4
**Confidence:** 4

**Summary:**

This paper proposed a new definition of Meta-Causal Graph as a world model, and used curiosity-driven intervention policy for causal discovery. They conduct experiments in chemical and magnetic tasks to verify.

From my perspective, the main contributions are two parts: 1. Use Meta-Causal Graph for causal discovery. 2. Design a curious reward fothe r exploration intervention. The proposed learning process is reasonable. But the experiments are not convincing.

**Questions:**

Please see the pros and cons.

**Ethical Concerns:**

["NO or VERY MINOR ethics concerns only"]

**Final Justification:**

The authors' rebuttal addressed my concerns. I suggest conducting more experiments in more challenging environments (Meta-World/CausalWorld) to demonstrate that the proposed method is a genuine meta-cause-and-effect world in the future. I maintain my positive score.

**Limitations:**

Please see the pros and cons.

**Quality:**

3

**Strengths And Weaknesses:**

Strengths:

1. Theoretically, they derive sufficient conditions ensuring that strategically chosen families of interventions uniquely identify meta-causal graphs, overcoming observational limitations.

2. They use meta meta-causal graph for causal discovery, and design a curious reward. The learning process is reasonable.

Weaknesses:

1. Need more experiments to verify the performance of the proposed method. Maybe other experimental environments can be conducted, not just chemical and manipulation. Moreover, you can revise the Magnetic tasks for generalization verification. The experiments in generalization are not enough.

2. Clear the words for the method description. In lines 51-56 of the method description, the learning process is not described clearly.

3. Explain the Meta-causal graph: from my understanding, the meta-causal graph is a combination of some graphs and their specific indicators. Right? For policy learning, learn those graphs and indicators, and then use them for downstream policy learning?

4. Explain the world. The author mentions that the method is a meta-causal world. I think the model learned in this work is a dynamics model, like CDL, Grader. It is not a world model.

5. More recent works should be discussed, for example:

        [1] Wang Z, Wang C, Xiao X, et al. Building minimal and reusable causal state abstractions for reinforcement learning[C]//Proceedings of the AAAI Conference on Artificial Intelligence. 2024, 38(14): 15778-15786.

        [2] Cao H, Feng F, Fang M, et al. Towards empowerment gain through causal structure learning in model-based reinforcement learning[C]//The Thirteenth International Conference on Learning Representations. 2025.

        [3] Yu Z, Ruan J, Xing D. Learning causal dynamics models in object-oriented environments[J]. arXiv preprint arXiv:2405.12615, 2024.

I hope that the author can explain the above questions.

---

> ### Author Rebuttal · Authors · 2025-07-30
>
> We thank the reviewer for highlighting our innovative Meta-Causal Graph framework, the effective curiosity-driven exploration strategy, and the strong theoretical guarantees for identifiability.
>
> > **W1: More environments & experiments in generalization.**
>
> Thank you for your suggestion. We **added new environment** and **adjusted magnetic force, density of ball and friction of the table** to test **the generalization ability** of our method. The results are shown below and the metric is epsiode reward on downstream tasks:
>
> |Reward↑|Small Magnetic Force ($\times 0.02$)|High Ball Density ($\times 10$)|Extra Table Friction|
> |-|-|-|-|
> |MLP|4.01|4.00|3.89|
> |FCDL|4.48|4.45|4.19|
> |**MCG**|**6.17**|**5.08**|**5.52**|
>
> We also **added the new experiment** to test the generalization ability of our method under varying levels ($n$) of noise (line 265-268) **when some variables are not observed**. The results are shown below and the metric is prediction accuracy:
>
> |Acc↑|Fork (n=2)|Fork (n=4)|Fork (n=6)|Chain (n=2)|Chain (n=4)|Chain (n=6)|
> |-|-|-|-|-|-|-|
> |FCDL|60.10|52.97|45.80|46.37|**47.88**|46.62|
> |MLP|31.56|31.67|31.93|25.95|24.71|21.66|
> |**MCG**|**66.94**|**59.19**|**46.35**|**49.93**|45.41|**48.09**|
>
> > **W2: Improve clarity of the method part.**
>
> Thank you for your suggestion. Here is a clearer summary of the learning process:
> - For each observed state, the agent predicts a **compact meta-causal graph (MCG)** representing the current context’s causal structure.
> - The agent selects actions based on a curiosity-driven reward, which encourages exploration of uncertain or unseen contexts.
> - The agent uses the learned MCG to model and **predict environment dynamics**.
> - As the agent collects new experience, both the meta-state assignment and the corresponding causal graphs are iteratively updated.
>
> We will improve the clarity of the method description in the revised version.
>
> > **W3: Explanation of the Meta-causal graph.**
>
> - Yes, your understanding is correct. The meta-causal graph is a collection of causal subgraphs, each associated with a specific meta state indicator.
> - We use the meta-causal graph for downstream tasks, where the agent learns to **predict the transition dynamics based on the meta state and its corresponding causal subgraph**.
> - Importantly, these meta states are **unknown** to the agent and must be discovered through interaction, which makes the problem challenging. The joint identifiability of both the causal subgraphs and the underlying meta states requires rigorous theoretical justification, as it is a fundamentally difficult problem.
>
> > **W4: Why is the Meta-Causal Graph considered a world model?**
>
> - The definition of a world model remains a subject of ongoing debate [Ding et al., 2024]. Generally, **a world model is any system that models the environment’s dynamics**—i.e., it learns the function mapping (state, action) pairs to next states. In this broad sense, CDL and Grader are also world models.
> - Our approach aligns with the broader definition of a world model: we first infer a compact **meta-causal graph** from each input, capturing the context-specific causal structure, and then use this representation to **predict future dynamics**.
> - This enables our model not only to predict transitions, like a dynamics model, but also to maintain an **abstract and interpretable understanding of the environment’s underlying structure**.
>
> > **W5: More recent work should be discussed.**
>
> - We appreciate your suggestion and give a brief comparison here. We will include more recent works and discuss comparisons in the revised version.
>     - [Wang et al., 2024]: This observation-based, passive learning method **cannot actively explore or adapt in open-ended worlds**. In contrast, our intervention-based approach **actively discovers unseen meta-states to learn a Meta-Causal Graph**, making it well-suited for open-ended environments.
>     - [Cao et al., 2025]: Their causal structure is **global and fixed**, not adapting to state or meta-state changes. Our framework explicitly models **multiple context-dependent causal graphs, allowing discovery of diverse causal mechanisms** across meta-states.
>     - [Yu et al., 2024]: OOCDM learns a **shared causal structure** for each object class through passive observation, **without context-specific graphs or active exploration**. Our method discovers **multiple context-specific graphs** via **active, curiosity-driven interventions**, enhancing exploration and identifiability.
>
> - Additionally, We **added new baselines**: causal discovery using transformer attention weights ("transformer") and the Jacobian matrices of MLP layers ("sandy-mixture") from [Pitis et al., 2020] and [Urpí et al., 2024], for comparison. The results are shown below. The metrics used are next-state prediction accuracy and the average episode reward achieved by model-based planning on downstream tasks.
>
> |Acc↑|Fork (n=2)|Fork (n=4)|Fork (n=6)|Chain (n=2)|Chain (n=4)|Chain (n=6)|
> |-|-|-|-|-|-|-|
> |transformer|24.45|22.49|21.41|29.62|30.37|29.78|
> |sandy-mixure|31.93|32.47|33.72|30.08|29.31|27.43|
> |**MCG**|**63.18**|**50.47**|**50.04**|**51.99**|**49.78**|**49.69**|
>
> |Reward↑|Fork (n=2)|Fork (n=4)|Fork (n=6)|Chain (n=2)|Chain (n=4)|Chain (n=6)|
> |-|-|-|-|-|-|-|
> |transformer|6.20|6.13|6.71|6.67|6.73|7.40|
> |sandy-mixure|6.48|6.10|6.50|7.03|6.73|7.17|
> |**MCG**|**14.65**|**14.06**|**13.28**|**13.82**|**12.49**|**12.45**|
>
> Pitis, Silviu, et al. Counterfactual data augmentation using locally factored dynamics. NeurIPS, 2020
>
> Urpí, Núria, et al. Causal action influence aware counterfactual data augmentation. arXiv, 2024.
>
> Yang Liu, et al. Learning world models with identifiable factorization. NeurIPS, 2023.
>
> Zizhao Wang, et al. Building minimal and reusable causal state abstractions for reinforcement learning. AAAI, 2024.
>
> Hongye Cao, et al. Towards empowerment gain through causal structure learning in model-based RL. arXiv, 2025.
>
> Yu Zhongwei, et al. Learning causal dynamics models in object-oriented environments. arXiv, 2024.
>
> Jingtao Ding, et al. Understanding world or predicting future? a comprehensive survey of world models. ACM Computing Surveys, 2024.

---

> ### Comment · Reviewer_KoPL · 2025-08-06
>
> Thank you for the detailed response. I think you should conduct more experiments in more challenging environments (Meta-World/CausalWorld) to demonstrate that the proposed method is a genuine meta-cause-and-effect world in the future. I maintain my positive score.

---

> ### Author Response · Authors · 2025-08-06
> **Thank you for maintaining on the positive side!**
>
> Dear reviewer KoPL,
>
> Thank you for maintaining a positive score and for suggesting Meta-World/CausalWorld. We agree they are compelling testbeds and view your pointer as a valuable direction beyond the present scope.
>
> Many thanks,
>
> The authors of #20694

---

### Note · Authors · 2025-08-14

Dear AC and all reviewers,

We sincerely appreciate the considerable time and effort you have dedicated in evaluating our work. To facilitate the AC-reviewer discussion, we summarize the up-to-date status as follows.

We are encouraged that three out of four reviewers were initially on the positive side, with scores of 5 (6jMv), 4 (KoPL), and 4 (n7bY).

**Reviewer 6jMv (initial score 5)**
- We thank Reviewer 6jMv in the final response ```I particularly appreciate the discussion on scalability and new experiment```, and recognized our unique contribution compared with Winn et al., ```Winn et al. is not RL-based and does not incorporate curiosity-driven mechanisms to encourage intervention-based exploration```. To increase impact of our work, we will follow the reviewer's suggestion to contextualize our work within broader MBRL/probabilistic graphical model literature - that is, we agree that there is meaningful overlap between _causal model and graphical models in RL_, which **we discussed in our rebuttal and last response**. We will definitely add these discussions to the final version.

**Reviewer KoPL (initial score 4)**
- We thank Reviewer KoPL in final note ```maintain my positive score``` and suggest adding experiments ```on more challenging environments Meta-World/CausalWorld```. From our experience with robotic environment in our paper (e.g., Magnetic), we realized such experiments are time-consuming, and thus not feasible to show results during limited discussion window (< 72 h). We will include results in our camera ready revision.

**Reviewer n7bY (initial score 4)**
- We are happy that Reviewer n7bY response in the final note ```I believe my concerns are addressed sufficiently``` - thank you so much!

For **Reviewer hzVP (unknown current upgraded score)**, who initially on the negative side, after discussion he/she wrote ```Thank you for the additional clarifications and the new experiments on different intrinsic objectives, this makes much more sense to me now. Truly appreciate your time and effort.``` and ```Thanks again, and I’ll proceed to update my rating.``` - we are so fortune to have such a responsible reviewer. The multi-round discussion has been deeply thought-provoking, and we will incorporate these useful suggestions to our final version.

We hope that our revisions and clarifications satisfactorily address the concerns raised. We thank AC and all reviewers again for your valuable time and expertise.

Many thanks,

Authors of #20694

---

### Decision · Program_Chairs · 2025-09-17

**Decision:**

Accept (poster)

**Comment:**

The paper proposes a method for learning a world model that adapts to changing causal dynamics. An agent actively explores using a curiosity-driven mechanism to find different situations (meta-states) and learns a separate graph of cause-and-effect relationships for each one. Reviewers acknowledge several strengths including theoretical contributions on identifiability, the importance of the problem setting, and the novelty of the curiosity-driven intervention approach. The weaknesses noted include: (1) attribution concerns regarding insufficient acknowledgment of FCDL, (2) insufficient experimental details and scope, (3) clarity issues for broader audiences, and (4) overstated claims about open-ended tasks and world model framing without sufficient empirical validation.

In the rebuttal, the authors provided new experiments, comparisons, and clarifications and acknowledged limitations regarding overstated claims. While three of four reviewers gave positive recommendations, key concerns remained unresolved regarding attribution, framing, and experimental scope limitations.

The Area Chair agrees with the majority of reviewers and recommends acceptance. Authors are advised to revise based on reviewers' comments, particularly by rewriting Section 4 to clearly distinguish novel contributions from prior work, moderating claims about world models and open-ended applications to match experimental scope, and improving accessibility for the broader ML community.